# F-Fidelity: A Robust Framework for Faithfulness Evaluation of Explainable AI

**Xu Zheng[1], Farhad Shirani[1] ✉ , Zhuomin Chen[1], Chaohao Lin[1], Wei Cheng [2],**
**Wenbo Guo[3], Dongsheng Luo[1] ✉**
[1]Florida International University, Miami, United States
[2]NEC Laboratories America, Princeton, United States
[3]University of California, Santa Barbara, United States
`{xzhen019,fshirani,zchen051,clin027,dluo}@fiu.edu`
`weicheng@nec-labs.com`
`henrygwb@ucsb.edu`

## Abstract

Recent research has developed a number of eXplainable AI (XAI) techniques, such as gradient-based approaches, input perturbation-base methods, and black-box explanation methods. While these XAI techniques can extract meaningful insights from deep learning models, how to properly evaluate them remains an open problem. The most widely used approach is to perturb or even remove what the XAI method considers to be the most important features in an input and observe the changes in the output prediction. This approach, although straightforward, suffers the Out-of-Distribution (OOD) problem as the perturbed samples may no longer follow the original data distribution. A recent method RemOve And Retrain (ROAR) solves the OOD issue by retraining the model with perturbed samples guided by explanations. However, using the model retrained based on XAI methods to evaluate these explainers may cause information leakage and thus lead to unfair comparisons. We propose Fine-tuned Fidelity (F-Fidelity), a robust evaluation framework for XAI, which utilizes i) an explanation-agnostic fine-tuning strategy, thus mitigating the information leakage issue, and ii) a random masking operation that ensures that the removal step does not generate an OOD input. We also design controlled experiments with state-of-the-art (SOTA) explainers and their degraded version to verify the correctness of our framework. We conduct experiments on multiple data modalities, such as images, time series, and natural language. The results demonstrate that F-Fidelity significantly improves upon prior evaluation metrics in recovering the ground-truth ranking of the explainers. Furthermore, we show both theoretically and empirically that, given a faithful explainer, F-Fidelity metric can be used to compute the sparsity of influential input components, i.e., to extract the true explanation size. The source code is available at https://trustai4s-lab.github.io/ffidelity.

## 1 Introduction

EXplainable AI (XAI) methods have been widely used in many domains, such as Computer Vision (CV) (Chattopadhay et al., 2018; Smilkov et al., 2017; Jiang et al., 2021; Zhou et al., 2016; Selvaraju et al., 2017), Neural Language Processing (NLP) (Lyu et al., 2024; Luo et al., 2024; Zhao et al., 2024), Graph Neural Networks (GNNs) (Ying et al., 2019; Luo et al., 2020; Yuan et al., 2020; Vu & Thai, 2020), and Time Series (Liu et al.; Queen et al., 2024). There are various types of explanation methods, in which the most predominant one is post-hoc instance-level explanation. Given a pre-trained classifier and a specific input, these methods aim to identify the most important features of the model's output. For instance, such explanations map to a subset of important pixels in image classification (Ribeiro et al., 2016; Selvaraju et al., 2017; Lundberg, 2017). Existing research has proposed a number of XAI methods to draw post-hoc explanations — such as Integrated Gradi-

---

✉Corresponding authors

ents (Sundararajan et al., 2017), CAM-based approaches (Selvaraju et al., 2017), and SmoothGrad (Smilkov et al., 2017).

Despite extracting useful insights about model decisions, how to faithfully evaluate and compare explanation methods remains an open challenge. There have been some existing efforts to address this issue. The preliminary works create datasets with "ground-truth" explanations for XAI evaluation, such as the MS-CoCo dataset in CV, (Lin et al., 2014), the Mutag dataset in graph (Debnath et al., 1991), and the e-SNLI dataset in NLP (Camburu et al., 2018). However, these datasets are still limited to certain tasks and cannot be generalized. More importantly, these "ground-truth" explanations are created based on humans' understanding of a data sample, which may not reflect an ML model's decision-making processes, and thus may still give unfaithful evaluations. More recent works propose the *removal strategy*, which does not rely on "ground-truth" explanations (Zhou et al., 2016; Selvaraju et al., 2017; Rong et al., 2022; Hooker et al., 2019; Madsen et al., 2023; 2022; Hase et al., 2021; Zheng et al., 2024; Yuan et al., 2022). Technically speaking, the removal strategy removes certain parts of an input that are deemed as (non-)important and records the changes in the model prediction. A larger drop in accuracy when removing the important parts reflects the removed parts as indeed important and thus validates the faithfulness of the corresponding explanation method. For example, in computer vision, removal means setting important pixels to zero/black; in natural language processing, it means replacing chosen tokens with "[MASK]" token or zero embeddings. Specific metrics designed based on the removal strategy include the Most Relevant First (MoRF) and Least Relevant First (LeRF) (Samek et al., 2016; Yuan et al., 2022) in CV and Importance Measures (Madsen et al., 2021; 2023) in NLP. However, these removal-based explanations suffer from the Out-Of-Distribution (OOD) issue (Hooker et al., 2019), as the perturbed samples with removed features may no longer follow the original distribution. As a result, the model predictions of the perturbed samples are unreliable, regardless of the explanation faithfulness.

To address the OOD problem, RemOve And Retrain (ROAR) (Hooker et al., 2019), proposed retraining the model for evaluating the explainer, where the model is trained using explanation-guided removal. However, retraining the model based on the explainer output leads to an information leakage issue (Rong et al., 2022). An additional issue, which has been overlooked in prior works, is that the removal strategy is highly dependent on explanation size ( or sparsity). That is, the size of the explanation part that is removed affects the evaluation output. In the absence of ground-truth explanations, it is not clear what fraction of the input elements should be removed as part of the explanation, which leads to inconsistent evaluations.

To address the aforementioned challenges, we introduce Fine-tuned Fidelity (F-Fidelity), a novel framework to evelute the performance of explainers. F-Fidelity leverages i) an explanation-agnostic fine-tuning strategy to prevent information leakage and ii) a controlled random masking operation to overcome the OOD issues observed in the application of prior evaluation metrics. The fine-tuning process employs stochastic masking operations, such as randomly dropping pixels in images, tokens in language, or time steps in time series data, to generate augmented training samples. This augmented data is then used to fine-tune a surrogate model. Unlike previous approaches such as ROAR (Hooker et al., 2019), our strategy effectively mitigates the risk of information leakage and label bias by using explanation-agnostic stochastic masks, while also offering improved efficiency as it eliminates the need to retrain the model for each individual explainer. During the evaluation phase, F-Fidelity implements a removal strategy that generates stochastic masks conditioned on the explainer output, designed to be in-distribution with respect to the masks used in the fine-tuning step, which mitigates the OOD issues observed in the application of prior evaluation metrics.

To verify the effectiveness of F-Fidelity, we provide comprehensive empirical faithfulness evaluations on a collection of explainers that are systematically degraded from an original explainer through controlled random perturbations. Thus the correct (ground-truth) ranking of the explainers, in terms of faithfulness, is known beforehand. The experiments on multiple data modalities demonstrate the robustness of F-Fidelity through both macro and micro correlations. Macro correlations measure the Spearman rank correlation between the ground truth and the overall performance across all sparsity levels, while micro correlations capture the average Spearman rank correlation between the ground truth and performance at individual sparsity levels. In both aspects, F-Fidelity consistently outperforms existing methods. Furthermore, we show both theoretically and empirically that, given a faithful explainer, the F-Fidelity metric can be used to compute the size (or sparsity) of ground-truth explanations. To elaborate, we show that the F-Fidelity metric output, when evaluated

a function of mask size, produces a piecewise constant function, where the length of the constant pieces depends on the explanation size. The following summarizes our main contributions:

- We introduce a novel evaluation framework for measuring explanation faithfulness with strong theoretical principles that is robust to distribution shifts.

- We design and implement a rigorous experimental setting to fairly compare metrics. Our comprehensive evaluations across multiple data modalities (images, time series, and natural language) demonstrate the superior performance and broad applicability of F-Fidelity.

- We theoretically and empirically analyze the relationship between the explanation size and the F-Fidelity metrics and demonstrate in F-Fidelity, the ground truth explanation sizes can be inferred.

## 2 PRELIMINARIES

### 2.1 NOTATION

Sets are denoted by calligraphic letters such as $\mathcal{X}$. The set $\{1, 2, \cdots, n\}$ is represented by $[n]$. Multidimensional arrays (tensors) are denoted by bold-face letters such as $\mathbf{x}$. Upper-case letters such as $X$ represent random variables, and lower-case letters such as $x$ represent their realizations. Similarly, random tensors are denoted by upper-case letters such as $\mathbf{X}$.

### 2.2 CLASSIFICATION MODELS AND EXPLANATION FUNCTIONS

Let $f : \mathbf{X} \mapsto Y$ be a (pre-trained) classification model — such as a neural network — which takes an input $\mathbf{X} \in \mathbb{R}^{t \times d}$ and outputs a label $Y \in \mathcal{Y}$, where $\mathcal{Y}$ is a finite set of labels. In CV tasks, $t = h \times w$, where $h, w$ are the height and width, and $d$ is the number of channels per input pixel. Analogously, in NLP and time series classification tasks, $t \in \mathbb{N}$ represents the time index, and $d$ is the feature dimension. An explanation function (explainer) consists of a pair of mappings $\psi = (\phi, \xi)$, where $\phi : \mathbf{X} \mapsto \mathbb{R}_+^{t \times d}$ is the score function, mapping each input element to its (non-negative) important score, and a mask function $\xi : \phi(\mathbf{X}) \mapsto \mathbf{M}$, mapping the output of the score function to a binary mask $\mathbf{M} \in \{0, 1\}^{t \times d}$. The masked input $\mathbf{X} \odot \mathbf{M}$ is called the explanation for the input $\mathbf{X}$ and model $f(\cdot)$, where $\odot$ represents elementwise multiplication. We denote the explanation size by $S = \|\mathbf{M}\|_1$, where $\| \cdot \|_1$ is the $\ell_1$ norm. That is, the explanation size $S$ is the number of non-zero elements of $\mathbf{M}$. In general, the size may be deterministically set to a constant value $s$, or alternatively, it may depend on the output of the score function, e.g., input elements receiving a score higher than a given threshold are included in the mask and the rest are removed. Let us assume that the (ground-truth) data distribution is $P_{\mathbf{X}, Y}$. Then, a 'good' explainer is one which minimizes the total variation distance $d_{TV}(P_{Y|\mathbf{X}}, P_{Y|\mathbf{X} \odot \mathbf{M}})$, while satisfying an explanation size constraint $\mathbb{E}_{\mathbf{X}}(\|\mathbf{M}\|_1) \leq s$, where $s \in \mathbb{N}$ is the desired average explanation size. The minimization of the total variation essentially enforces that the posterior distribution of the classifier output be mostly determined by the masked input explanation, implying that the subset of input components which are removed by the mask have a low influence on the classifier output.

### 2.3 QUANTIFYING THE PERFORMANCE OF EXPLAINERS

A key challenge in explainability research is to quantify and compare the performance of various explainers. The performance of an explainer can be formally quantified in terms the total variation distance $d_{TV}(P_{Y|\mathbf{X}}, P_{Y|\mathbf{X} \odot \mathbf{M}})$ as a function of the average explanation size $\mathbb{E}_{\mathbf{X}}(\|\mathbf{M}\|_1)$. However, in most problems of interest, the underlying statistics $P_{\mathbf{X}, Y}$ is not available, and hence direct evaluation of the aforementioned total variation distance is not possible. As discussed in the introduction, some datasets are accompanied by ground-truth explanations, which enables the use of measures such as AUC and IoU for evaluating the quality explainers. However, the ground-truth explanations are available only for a limited collection of datasets, and even when ground-truth explanations are available, they may not accurately reflect the model's internal decision-making processes. To address the aforementioned issues, a widely used set of metrics have been proposed in the literature which are based on the removal strategy. In CV, two removal orders have been considered (Samek et al., 2016; Yuan et al., 2022): MoRF, which evaluates explanations by removing the most influential pixels first, and LeRF, which begins with removing the least influential pixels. These approaches

provide complementary perspectives on feature importance - MoRF assesses whether removing important features significantly impacts predictions, while LeRF verifies if retaining important features is sufficient for model performance. In the graph domain, an alternative but conceptually related removal strategy is used, where first the size of explanation subgraphs is determined according to a sparsity parameter, and then edges are removed either from the explanation subgraph of the desired size, or the non-explanation subgraph which is its complement. In this paper, we use metrics based on this graph domain removal strategy, called the *Fidelity* (Fid) metric in the literature (Pope et al., 2019; Bajaj et al., 2021; Yuan et al., 2022). The Fidelity metrics align with MoRF and LeRF principles through $Fid^+$ and $Fid^-$ respectively. Formally, given the input and label pair $(\mathbf{x}, y)$ and a binary mask $\mathbf{m}$, the $Fidelity$ metrics are defined as follows:

$$Fid^+(\psi) = \frac{1}{|\mathcal{T}|} \sum_{(\mathbf{x}_i, y_i) \in \mathcal{T}} \mathbb{1}(y_i = f(\mathbf{x}_i)) - \mathbb{1}(y_i = f(\mathbf{x}_i - \mathbf{x}_i \odot \mathbf{m}_i)) \tag{1}$$

$$Fid^-(\psi) = \frac{1}{|\mathcal{T}|} \sum_{(\mathbf{x}_i, y_i) \in \mathcal{T}} \mathbb{1}(y_i = f(\mathbf{x}_i)) - \mathbb{1}(y_i = f(\mathbf{x}_i \odot \mathbf{m}_i)), \tag{2}$$

where $\mathcal{T}$ is the dataset used for evaluating the performance of the explainer, $n$ is the size of the dataset, $\mathbb{1}(\cdot)$ denotes the indicator function, and $\mathbf{m}_i = \psi(\mathbf{x}_i)$ is the explanation corresponding to $\mathbf{x}_i$ produced by the explainer $\psi(\cdot)$. Here, $Fid^+$ measures prediction changes when removing important features (similar to MoRF), while $Fid^-$ evaluates model performance when keeping only important features (similar to LeRF).

A significant limitation of explanation evaluation through removal strategies is the OOD problem (Hooker et al., 2019; Zheng et al., 2024). When we remove elements from an input - whether they are pixels in images, time steps in time series, or edges in graphs - the modified input may no longer follow the original data distribution that the model was trained on. For instance, when evaluating image explanations by zeroing out important pixels, the resulting images with black patches are unlikely to resemble natural images. Consequently, the model's predictions on these modified inputs may be unreliable, not due to low quality of the explanation itself, but because the model is operating outside its training distribution.

This OOD problem has been analyzed through different approaches. In ROAR (Hooker et al., 2019), which focused on CV tasks, the solution was to retrain the model on perturbed data where important input elements identified by the explainer are removed. While this approach mitigates the OOD issue, Rong et al. (2022) identified that it suffers from information leakage through the binary masks of removed pixels - the pattern of removals itself can contain enough class information to affect the evaluation outcome. Moreover, ROAR requires retraining a separate model for each explainer being evaluated, making it computationally expensive and time-consuming when comparing multiple explanation methods.

Similarly, In R-Fidelity (Zheng et al., 2024), which considered the evaluation of GNN explainers, it was argued that the Fidelity metric highly relies on the robustness of the underlying classifier to removal of potentially large sections of the input, e.g.,the removal of a large subgraph explanation for $Fid^+$ or its complement for $Fid^-$. That is, the classifier should be robust to OOD inputs for the Fidelity metric to align with those of the (theoretically justified) total-variation-based metric discussed in the previous sections. Rather than retraining the model, R-Fidelity introduces a stochastic removal strategy that addresses the OOD issue by controlling the size of removed sections and randomly sampling which elements to remove, thus limiting the distribution shift of perturbed inputs. Specifically, the following *Robust Fidelity metrics* (RFid) was introduced:

$$RFid^+(\psi, \alpha^+, s) = \frac{1}{|\mathcal{T}|} \sum_{(\mathbf{x}_i, y_i) \in \mathcal{T}} \mathbb{1}(y_i = f(\mathbf{x}_i)) - P(y_i = f(\chi^+(\mathbf{x}_i, \alpha^+, s)) \tag{3}$$

$$RFid^-(\psi, \alpha^-, s) = \frac{1}{|\mathcal{T}|} \sum_{(\mathbf{x}_i, y_i) \in \mathcal{T}} \mathbb{1}(y_i = f(\mathbf{x}_i)) - P(y_i = f(\chi^-(\mathbf{x}_i, \alpha^-, s)), \tag{4}$$

where $\chi^+(\mathbf{x}_i, \alpha^+, s)$ is a sampling function which randomly, uniformly, and independently removes $\lfloor s\alpha^+ \rceil$ elements from the $s$ highest scoring elements of $\mathbf{x}_i$ based on the scores produced by $\phi(\mathbf{x}_i)$, and $\chi^-(\mathbf{x}_i, \alpha^-)$ removes $\lceil (td - s)\alpha^- \rceil$ elements from the lowest scoring $td - s$ elements. If $\alpha^+ = \alpha^- = 1$, then the RFid metric reduces to the Fid metric. On the other hand, as $\alpha^+$ and $\alpha^-$ are

decreased, fewer input elements are removed, hence requiring lower OOD robustness to ensure the accuracy of the evaluation output.

## 3 ROBUST FIDELITY VIA FINE-TUNING AND STOCHASTIC REMOVAL

As discussed in the previous section, a significant limitation of prior explanation evaluation metrics is the loss in accuracy due to the OOD nature of the modified inputs generated by the application of removal strategies. For instance, the probability difference $P(Y = f(\mathbf{X})) - P(Y = f(\mathbf{X} - \mathbf{X} \odot \mathbf{M}))$ may be large, even for low-quality explanations. This occurs because the modified input $\mathbf{X} - \mathbf{X} \odot \mathbf{M}$ is OOD for the trained classifier $f(\cdot)$, despite $P_{Y|\mathbf{X}}$ and $P_{Y|\mathbf{X} - \mathbf{X} \odot \mathbf{M}}$ being close to each other. Consequently, this yields a high $Fid^+$ score despite the explanation's low quality with respect to the theoretically justified total variation metric. For example, in the empirical evaluations provided in the next sections, we demonstrate that the Fid measure sometimes assigns better evaluations to completely random explainers than to those whose outputs align with ground-truth explanations.

A partial solution in the graph domain addresses this issue by removing only an $\alpha^+$ fraction of the explanation subgraph and $\alpha^-$ fraction of the non-explanation subgraphs (Zheng et al., 2024). However, we argue that two issues degrade the evaluation quality of the RFid metric. First, the classifier may lack robustness and produce unreliable outputs even when the input is only slightly perturbed. Second, if the original explanation size is large (small), then removing an $\alpha^+$ ($\alpha^-$) of the explanation (non-explanation) part of the input, this would still yield OOD inputs.

In this work, we introduce a simple yet effective framework, F-Fidelity, for robust evaluation of XAI methods. Our strategy can be summarized as **fine-tuning and stochastic removal**. Specifically, we first fine-tune the model with randomly masked inputs to improve its robustness to perturbation. Then employ a controlled stochastic removal process that ensures the perturbed inputs remain within the distribution seen during fine-tuning.

To achieve reliable predictions on partially removed inputs, we design a fine-tuning process that randomly removes up to $\beta \in [0, 1]$ ratio of input elements. To elaborate, we introduce a stochastic mask generator $P_\beta : (t, d) \mapsto \mathbf{M}_\beta$, which takes the input dimensions $(t, d)$ as input and outputs a mask $\mathbf{M}_\beta \in \{0, 1\}^{t \times d}$ of *size $\beta t d$*, i.e. with up to $\beta t d$ non-zero elements. For instance, in image classification, the mask generator is designed to select random image pixels or patches for removal. Formally, we define the fine-tuning loss as:

$$L = \mathbb{E}_{\mathbf{X}, Y} \left[ \mathcal{L} \left( f \left( \mathbf{X} - P_\beta \odot \mathbf{X} \right), Y \right) \right], \tag{5}$$

where $\mathcal{L}$ is the loss function used during training (e.g., cross-entropy).

In the evaluation process, we modify the RFid metric to ensure consistency with our fine-tuning strategy by upper-bounding the total number of removed elements by $\beta t d$ — the same bound used during fine-tuning. That is, for a fixed $\beta$, and RFid parameters $\alpha^+_{orig}, \alpha^-_{orig} \in [0, 1]$, we set the upper-bounded RFid parameters as

$$\alpha^+ = \min(\alpha^+_{orig}, \frac{\beta t d}{s}) \quad \text{and} \quad \alpha^- = \min(\alpha^-_{orig}, \frac{\beta t d}{(td - s)}), \tag{6}$$

so that the sampling functions $\chi^+$ and $\chi^-$ remove the minimum of $\alpha^+_{orig} s$ (based on explanation size) and $\beta t d$ (based on input size) elements for $\chi^+$, and the minimum of $\alpha^-_{orig}(td - s)$ and $\beta t d$ elements for $\chi^-$, providing absolute upper bounds on the number of removed elements.

The pipeline of our method is shown in Algorithm 1. We denote the resulting metrics, which use the fine-tuning process and the $RFid^+$ and $RFid^-$ metrics with sampling rates that are truncated based on $\beta$ (equation 6), as $FFid^+(\psi, \alpha^+_{orig}, \beta, s)$ and $FFid^-(\psi, \alpha^-_{orig}, \beta, s)$, respectively.

It is worth noting that both $FFid^+$ and $FFid^-$ can take negative values in certain cases. This occurs when the accuracy after masking exceeds the original prediction accuracy. Alternative formulations could enforce positive values by only reporting masked accuracy like ROAR (Hooker et al., 2019) and deletion/insertion scores (Petsiuk et al., 2018; Pan et al., 2021).

# 4 DETERMINING THE EXPLANATION SIZE VIA FIDELITY METRICS

A critical challenge in explainable AI is determining the appropriate size or scope of explanations. Ideally, ground truth explanations would be discretized into distinct clusters representing different levels of importance. For instance, in image classification, pixels associated with the target object tend to receive high importance scores. Conversely, pixels corresponding to the background or irrelevant regions receive low scores. However, in many practical scenarios, even *good* explainers that produce accurate explanation masks — as measured by the Fid and RFid evaluation metrics — may yield explanation scores that are not discretized into distinct clusters.

In this section, we theoretically demonstrate that our proposed evaluation metric can recover the cluster sizes given an explainer that outputs the correct explanation mask (i.e., correctly ranks the importance of input elements). Thus, provided the explainer outputs an accurate mask function, our metric can recover the explanation size (also known as sparsity). To provide a concrete theoretical analysis, we first consider a classification problem under a set of idealized assumptions that generalize the above observations on the clustering of explanation scores. Specifically, let us consider a classification task defined by a joint distribution $P_{\mathbf{X},Y}$ and a classifier $f : \mathbf{x} \mapsto y$. We assume that the input elements can be partitioned into several *influence tiers*. That is, for any given input $\mathbf{x}$, there exists a partition $\mathcal{C}_k(\mathbf{x}), k \in [r]$ of the index set $[t] \times [d]$, where $\mathcal{C}_k(\mathbf{x})$ represents the set of indices of the input elements belonging to tier $k$, and $c_k = |\mathcal{C}_k(\mathbf{x})|$ are the (fixed) tier sizes. For a given mask $\mathbf{m}$, the probability of correct classification based on the masked input $\mathbf{x} \odot \mathbf{m}$ depends only on the counts of unmasked elements in each influence tier. Formally, $P(Y|\mathbf{x} \odot \mathbf{m}) = g(j_1, j_2, \ldots, j_r)$, where $g : [c_1] \times [c_2] \times \cdots \times [c_r] \to [0, 1]$ is a function monotonically increasing with respect to the lexicographic ordering on its input, and $j_k \in [c_k]$ is the number of elements in $\mathcal{C}_k(\mathbf{x})$ whose corresponding mask element in $\mathbf{m}$ is non-zero (unmasked). We further focus on Shapley-value-based explanations, which provide a theoretical foundation for our analysis. Recall that, given label $y$, the Shapley value associated with an element $(i, j) \in [t] \times [d]$ of $\mathbf{x}$ is given as (Lundberg, 2017):

$$S_{\mathbf{x}}(i, j) = \sum_{\mathbf{m}:m_{i,j}=0} \frac{\|m\|_1!(td - \|m\|_1 - 1)!}{(td)!} (P(Y = y|\mathbf{x} \odot \mathbf{m}') - P(Y = y|\mathbf{x} \odot \mathbf{m})),$$

where $\mathbf{m}'$ is the mask obtained from $\mathbf{m}$ by setting $m'_{i,j} = 1$ (unmasking the $(i, j)$ element). Under the aforementioned influence tier assumption, it is straightforward to verify that input elements within the same influence tier receive equal Shapley values. Specifically, for any $k \in [r]$ and any $(i, j), (i', j') \in \mathcal{C}_k(\mathbf{x})$, we have $S_{\mathbf{x}}(i, j) = S_{\mathbf{x}}(i', j')$.

**Theorem 1.** *For the classification task described above, and a given pre-trained classifier $f(\cdot)$, consider a Shapley-value-based explainer $\psi(\cdot)$. For $\alpha_{orig}^+ \in [0, 1]$ and $\beta \in [0, \alpha^+]$, let*

$$e(s) = \mathbb{E}_{\mathbf{X},Y}(FFid^+(\psi, \alpha_{orig}^+, \beta, s)), \quad s \in [0, td].$$

*Then, $e(s)$ is monotonically increasing for $s \in [0, c_1]$ and monotonically decreasing for $s \in [\max(\frac{\beta}{\alpha_{orig}^+} td, c_1), td]$.*

The proof is provided in the Appendix.

This theorem shows that $FFid^+$ can recover the size of the most influential tier (i.e., the first cluster size) when the explainer's ranking is close to that of an ideal Shapley-based explainer. Specifically, the value of $s$ in which $FFid^+$ changes direction corresponds to the size of an influence tier. This is verified in the empirical evaluation provided in Section D.8. This result implies that even when an explainer provides continuous scores without distinct clustering, our metric can infer the underlying discrete structure of the ground truth explanations and recover the explanation size.

# 5 EXPERIMENTS

To demonstrate the robustness of the F-Fidelity framework, we conduct comprehensive experiments across multiple domains, including image classification, time series analysis, and natural language processing. Our evaluation strategy builds upon the concept introduced by Rong et al. (2022), which posits that an ideal evaluation method should yield consistent rankings in both MoRF and LeRF settings. To further establish a controlled experimental setting with ground truth (GT) rankings, we

introduce a novel approach for a fair comparison using a degradation operation on a good explanation, such as an explanation obtained by Integrated Gradients (IG) (Sundararajan et al., 2017), generating a series of explanations with varying levels of random noise. Specifically, we first obtain initial (good) explanations using well-established explainers, then systematically degrade them by adding different ratios of random noise. Since explanation quality naturally decreases with increased noise, this creates a ground truth ranking where explanations with less noise should rank higher.

We evaluate the performance of F-Fidelity against established baselines such as (Fidelity), ROAR (Hooker et al., 2019), and R-Fidelity (Zheng et al., 2024) across a wide range of sparsity levels, from 5% to 95% at 5% intervals. Throughout our evaluation, we focus on three key Spearman rank correlations that measure how well different evaluation metrics align with ground truth and each other. Specifically, "MoRF vs. GT" measures how well $Fid^+$ rankings align with ground truth rankings when removing important features first, while "LeRF vs. GT" measures the correlation between $Fid^-$ rankings and ground truth when retaining important features. The "MoRF vs. LeRF" correlation assesses the consistency between $Fid^+$ and $Fid^-$ evaluations, where strong negative correlation indicates that features identified as important by one metric are consistently identified as important by the other. These correlations allow us to assess the methods' performance under various conditions, from highly sparse to nearly complete explanations. To provide a thorough analysis, we employ both macro and micro correlation metrics:

- **Macro Correlation**: Following Rong et al. (2022)'s approach of evaluating overall explainer consistency, and inspired by the AUC-based aggregation methods in Zhu et al. (2024) and Pan et al. (2021), we compute the AUC with respect to sparsity across the entire 5-95% range for each explanation method. The macro correlations are then calculated using these AUC values, providing an overall performance measure across all sparsity levels.
- **Micro Correlation**: To capture fine-grained performance differences, we calculate micro correlations at each sparsity level. In the main body of the paper, we report the averaged micro correlations across all sparsity levels, as well as the average rank of each method.

### 5.1 IMAGE CLASSIFICATION EXPLANATION EVALUATION

**Setup**. We use CIFAR-100 (Krizhevsky et al., 2009) and Tiny-Imagenet (Deng et al., 2009)[1], as the benchmark datasets. To obtain a pre-trained model to be explained, we adopt ResNet (He et al., 2016). More experiments with Vision Transformer(ViT) (Dosovitskiy et al., 2020) can be found in Appendix D.1. To generate different explanations, we first use two explanation methods, SmoothGrad Squared (SG-SQ) (Smilkov et al., 2017) and GradCAM (Selvaraju et al., 2017) to obtain the explanations. Then we use a set proportion of noise perturbations in the image, $[0.0, 0.2, 0.4, 0.6, 0.8, 1.0]$. We provide the implementation detail in Appendix C.

**Results**. As shown in Table 1 and Table 2, F-Fidelity achieves superior performances across different explainers and correlation metrics. Traditional Fidelity and ROAR methods show inconsistent and often poor performance, particularly in MoRF scenarios. Fidelity suffers from the out-of-distribution (OOD) issue, leading to unreliable evaluations when features are removed. ROAR, while addressing the OOD problem through retraining, faces challenges with information leakage and potential convergence issues, resulting in suboptimal correlations. In contrast, for both SG-SQ and GradCAM, F-Fidelity consistently achieves optimal or near-optimal performance in ranking explanations. In CIFAR-100, F-Fidelity achieves perfect Macro and Micro correlations for all three cases with SG-SQ. For GradCAM, it shows strong negative correlation (-0.60 to -0.71) in MoRF comparisons, significantly outperforming other methods. Tiny ImageNet results further reinforce its effectiveness with perfect correlations (-1.00) across all metrics for both explainers. Notably, F-Fidelity consistently ranks first in the micro rank evaluation for "MoRF vs. GT" and "MoRF vs. LeRF" correlations across both datasets and explainers, indicating its robust performance across various sparsity levels.

### 5.2 TIME SERIES CLASSIFICATION EXPLANATION EVALUATION

**Setup.** We use two benchmark datasets for time series analysis: PAM for human activity recognition and Boiler for mechanical fault detection (Queen, 2023). For PAM, we use 534 samples across 8

---

[1]https://github.com/rmccorm4/Tiny-Imagenet-200?tab=readme-ov-file

Table 1: Spearman rank correlation and rank results on CIFAR-100 dataset with ResNet.

| | Correlation | SG-SQ | | | | GradCam | | | |
|---|---|---|---|---|---|---|---|---|---|
| | | Fidelity | ROAR | R-Fidelity | F-Fidelity | Fidelity | ROAR | R-Fidelity | F-Fidelity |
| **Macro Corr.** | MoRF vs GT ↓ | -0.68±0.08 | -0.66±0.00 | **-1.00**±0.00 | **-1.00**±0.00 | 0.81±0.03 | 0.99±0.02 | 0.20±0.11 | **-0.60**±0.00 |
| | LeRF vs GT ↑ | **1.00**±0.00 | **1.00**±0.00 | **1.00**±0.00 | **1.00**±0.00 | **1.00**±0.00 | **1.00**±0.00 | **1.00**±0.00 | **1.00**±0.00 |
| | MoRF vs LeRF ↓ | -0.68±0.08 | -0.66±0.00 | **-1.00**±0.00 | **-1.00**±0.00 | 0.81±0.03 | 0.99±0.02 | 0.20±0.11 | **-0.60**±0.00 |
| **Micro Corr.** | MoRF vs GT ↓ | -0.36±0.37 | -0.54±0.35 | **-1.00**±0.01 | **-1.00**±0.00 | 0.57±0.14 | 0.76±0.07 | 0.09±0.12 | **-0.71**±0.02 |
| | LeRF vs GT ↑ | 0.98±0.04 | 0.99±0.04 | **1.00**±0.00 | **1.00**±0.00 | 0.99±0.01 | 0.99±0.01 | **1.00**±0.00 | **1.00**±0.00 |
| | MoRF vs LeRF ↓ | -0.23±0.35 | -0.53±0.34 | **-1.00**±0.01 | **-1.00**±0.00 | 0.57±0.14 | 0.75±0.07 | 0.01±0.12 | **-1.00**±0.02 |
| **Micro Rank** | MoRF vs GT ↓ | 3.68±0.46 | 3.26±0.44 | 1.11±0.31 | **1.00**±0.00 | 3.00±0.46 | 3.74±0.55 | 2.11±0.79 | **1.00**±0.00 |
| | LeRF vs GT ↓ | **1.11**±0.31 | 1.21±0.41 | 1.42±0.67 | 1.42±0.67 | 2.05±1.39 | 1.68±1.08 | 1.21±0.69 | **1.11**±0.45 |
| | MoRF vs LeRF ↓ | 3.68±0460 | 3.32±0.46 | 1.11±0.31 | **1.00**±0.00 | 3.00±0.45 | 3.74±0.55 | 2.16±0.74 | **1.00**±0.00 |

Table 2: Spearman rank correlation results on Tiny-Imagenet dataset with ResNet.

| | Correlation | SG-SQ | | | | GradCam | | | |
|---|---|---|---|---|---|---|---|---|---|
| | | Fidelity | ROAR | R-Fidelity | F-Fidelity | Fidelity | ROAR | R-Fidelity | F-Fidelity |
| **Macro Corr.** | MoRF vs GT ↓ | -0.69±0.07 | -0.83±0.00 | **-1.00**±0.00 | **-1.00**±0.00 | -0.97±0.03 | 0.63±0.03 | **-1.00**±0.00 | **-1.00**±0.00 |
| | LeRF vs GT ↑ | **1.00**±0.00 | **1.00**±0.00 | **1.00**±0.00 | **1.00**±0.00 | **1.00**±0.00 | **1.00**±0.03 | **1.00**±0.00 | **1.00**±0.00 |
| | MoRF vs LeRF ↓ | -0.69±0.07 | -0.83±0.00 | **-1.00**±0.00 | **-1.00**±0.00 | -0.97±0.03 | 0.63±0.03 | **-1.00**±0.00 | **-1.00**±0.00 |
| **Micro Corr.** | MoRF vs GT ↓ | -0.38±0.48 | -0.50±0.37 | -0.99±0.03 | **-1.00**±0.00 | -0.42±0.14 | 0.54±0.14 | -0.99±0.01 | **-1.00**±0.00 |
| | LeRF vs GT ↑ | **1.00**±0.00 | **1.00**±0.01 | **1.00**±0.00 | **1.00**±0.00 | **1.00**±0.00 | 0.99±0.00 | **1.00**±0.00 | **1.00**±0.00 |
| | MoRF vs LeRF ↓ | -0.38±0.48 | -0.50±0.36 | -0.99±0.03 | **-1.00**±0.01 | -0.42±0.14 | 0.55±0.16 | -0.99±0.01 | **-1.00**±0.00 |
| **Micro Rank** | MoRF vs GT ↓ | 3.74±0.44 | 3.21±0.41 | 1.16±0.36 | **1.00**±0.00 | 3.05±0.51 | 3.84±0.36 | 1.21±0.41 | **1.00**±0.00 |
| | LeRF vs GT ↓ | 1.11±0.31 | **1.00**±0.00 | 1.16±0.49 | 1.16±0.49 | 1.47±0.94 | 1.58±1.04 | 1.16±0.67 | **1.11**±0.44 |
| | MoRF vs LeRF↓ | 3.74±0.44 | 3.21±0.41 | 1.10±0.31 | **1.00**±0.00 | 3.16±0.36 | 3.84±0.36 | 1.21±0.41 | **1.00**±0.00 |

activity classes, with each sample recorded using a fixed segment window length of 600 from 17 sensors. The Boiler dataset, used for mechanical fault detection, consists of 400 samples with 20 dimensions and a fixed segment window length of 36. We employ IG from the Captum library [2] to obtain initial explanations. To generate different explanations, we apply noise perturbations to the importance of each timestamp, using proportions of $[0.0, 0.1, 0.2, 0.3, 0.4, 0.5]$ for PAM and $[0.0, 0.2, 0.4, 0.6, 0.8, 1.0]$ for Boiler.

**Results.** Table 3 demonstrates the robust performance of F-Fidelity across different evaluation metrics. Specifically, in the PAM dataset, F-Fidelity and R-Fidelity outperform Fidelity and ROAR in micro correlations and ranks, with F-Fidelity slightly edging out R-Fidelity in "LeRF vs GT" and "MoRF vs LeRF" micro ranks. The Boiler dataset results reveal more significant differences among the methods. F-Fidelity substantially outperforms other methods in "LeRF vs GT" and "MoRF vs LeRF" macro correlations. In micro correlations and ranks for the Boiler dataset, F-Fidelity consistently achieves the best or near-best performance across all metrics. These results underscore the effectiveness and stability of F-Fidelity in evaluating explanations for time series data.

Table 3: Spearman ranks correlations and ranks on time series datasets with LSTM as classifier. The best performance is marked as bold. the "-" means the correlation can't be obtained because of the same rank of different explanations.

| | Correlation | PAM | | | | Boiler | | | |
|---|---|---|---|---|---|---|---|---|---|
| | | Fidelity | ROAR | R-Fidelity | F-Fidelity | Fidelity | ROAR | R-Fidelity | F-Fidelity |
| **Macro Corr.** | MoRF vs GT ↓ | **-1.00**±0.00 | **-1.00**±0.00 | **-1.00**±0.00 | **-1.00**±0.00 | -0.98±0.03 | -0.99±0.20 | **-1.00**±0.00 | **-1.00**±0.00 |
| | LeRF vs GT ↑ | **1.00**±0.00 | **1.00**±0.00 | **1.00**±0.00 | **1.00**±0.00 | 0.22±0.03 | 0.36±0.11 | 0.53±0.06 | **0.86**±0.07 |
| | MoRF vs LeRF ↓ | **-1.00**±0.00 | **-1.00**±0.00 | **-1.00**±0.00 | **-1.00**±0.00 | -0.27±0.08 | -0.38±0.14 | -0.53±0.06 | **-0.86**±0.07 |
| **Micro Corr.** | MoRF vs GT ↓ | -0.88±0.39 | -0.97±0.09 | **-1.00**±0.00 | **-1.00**±0.00 | - | - | **-0.79**±0.31 | -0.78±0.29 |
| | LeRF vs GT ↑ | 0.74±0.33 | 0.80±0.21 | **1.00**±0.01 | **1.00**±0.00 | - | - | 0.69±0.35 | **0.79**±0.27 |
| | MoRF vs LeRF ↓ | -0.65±0.45 | -0.77±0.21 | **-1.00**±0.01 | **-1.00**±0.00 | - | - | -0.81±0.27 | **-0.97**±0.03 |
| **Micro Rank** | MoRF vs GT ↓ | 1.53±0.82 | 1.37±0.58 | **1.00**±0.00 | 1.05±0.22 | 2.95±0.51 | 2.73±0.71 | **1.21**±0.52 | 1.52±0.50 |
| | LeRF vs GT ↓ | 2.32±1.09 | 2.37±0.74 | 1.16±0.36 | **1.05**±0.22 | 2.59±0.99 | 2.74±0.64 | 2.21±1.00 | **1.68**±0.86 |
| | MoRF vs LeRF ↓ | 2.37±1.09 | 2.42±0.82 | 1.16±0.36 | **1.11**±0.31 | 2.95±0.22 | 2.95±0.22 | 1.89±0.31 | **1.05**±0.22 |

---

[2]https://github.com/pytorch/captum

## 5.3 Natural Language Classification Explanation Evaluation

**Setup.** We use two benchmark datasets for our NLP experiments: the Stanford Sentiment Treebank (SST2) (Socher et al., 2013) for binary sentiment classification and the Boolean Questions (BoolQ) (Socher et al., 2013) dataset for question-answering tasks. For SST2, we utilize 67,349 sentences for training and 872 for testing. BoolQ comprises 9,427 question-answer pairs for training and 3,270 for testing. We employ two popular model architectures: LSTM networks and Transformer-based models (Appendix D.1). We follow the setting in image classification. Specifically, to generate explanations, we first use IG to obtain initial explanations, then apply noise perturbations to the importance of each timestamp at levels of $[0.0, 0.2, 0.4, 0.6, 0.8, 1.0]$. We compare our F-Fidelity method against baselines including Fidelity and R-Fidelity, evaluating performance using both macro and micro Spearman correlations. We use the Adam optimizer (Kingma & Ba, 2015) with a learning rate of 1e-4, keeping other hyperparameters at their default values.

Table 4: Spearman rank correlation and rank results with LSTM model on SST2 and BoolQ datasets.

| Correlation | | SST2 | | | BoolQ | | |
|---|---|---|---|---|---|---|---|
| | | Fidelity | R-Fidelity | F-Fidelity | Fidelity | R-Fidelity | F-Fidelity |
| Macro Corr. | MoRF vs GT ↓ | $-1.00_{\pm 0.00}$ | $-1.00_{\pm 0.00}$ | $-1.00_{\pm 0.00}$ | $-1.00_{\pm 0.00}$ | $-1.00_{\pm 0.00}$ | $-1.00_{\pm 0.00}$ |
| | LeRF vs GT ↑ | $1.00_{\pm 0.00}$ | $1.00_{\pm 0.00}$ | $1.00_{\pm 0.00}$ | $1.00_{\pm 0.00}$ | $1.00_{\pm 0.00}$ | $1.00_{\pm 0.00}$ |
| | MoRF vs LeRF ↓ | $-1.00_{\pm 0.00}$ | $-1.00_{\pm 0.00}$ | $-1.00_{\pm 0.00}$ | $-1.00_{\pm 0.00}$ | $-1.00_{\pm 0.00}$ | $-1.00_{\pm 0.00}$ |
| Micro Corr. | MoRF vs GT ↓ | $-1.00_{\pm 0.01}$ | $-1.00_{\pm 0.01}$ | $-0.99_{\pm 0.03}$ | $-1.00_{\pm 0.01}$ | $-1.00_{\pm 0.01}$ | $-1.00_{\pm 0.00}$ |
| | LeRF vs GT ↑ | $1.00_{\pm 0.00}$ | $1.00_{\pm 0.00}$ | $1.00_{\pm 0.01}$ | $1.00_{\pm 0.00}$ | $1.00_{\pm 0.00}$ | $1.00_{\pm 0.00}$ |
| | MoRF vs LeRF ↓ | $-1.00_{\pm 0.01}$ | $-1.00_{\pm 0.01}$ | $-0.99_{\pm 0.02}$ | $-1.00_{\pm 0.01}$ | $-1.00_{\pm 0.01}$ | $-1.00_{\pm 0.00}$ |
| Micro Rank | MoRF vs GT ↓ | $1.00_{\pm 0.00}$ | $1.00_{\pm 0.00}$ | $1.05_{\pm 0.22}$ | $1.00_{\pm 0.00}$ | $1.00_{\pm 0.00}$ | $1.00_{\pm 0.00}$ |
| | LeRF vs GT ↓ | $1.05_{\pm 0.22}$ | $1.05_{\pm 0.22}$ | $1.05_{\pm 0.22}$ | $1.00_{\pm 0.00}$ | $1.00_{\pm 0.00}$ | $1.00_{\pm 0.00}$ |
| | MoRF vs LeRF ↓ | $1.00_{\pm 0.00}$ | $1.00_{\pm 0.00}$ | $1.11_{\pm 0.31}$ | $1.00_{\pm 0.00}$ | $1.00_{\pm 0.00}$ | $1.00_{\pm 0.00}$ |

**Results.** The results are shown in Table 3. Our experiments in the NLP domain reveal an interesting phenomenon that NLP models demonstrate remarkable robustness to OOD inputs, which is reflected in the performance of various fidelity metrics. Notably, the vanilla Fidelity (Fidelity) method achieves excellent results, often matching or closely approaching the performance of more sophisticated methods like R-Fidelity and our proposed F-Fidelity. This is evident from the near-perfect correlations (-1.00 or 1.00) across most metrics in both datasets. This strong performance of vanilla fidelity suggests that the OOD problem, which often necessitates more complex evaluation frameworks in other domains, may be less pronounced in NLP tasks. The robustness of NLP models to perturbations in input tokens likely contributes to this phenomenon, allowing simpler evaluation methods to maintain their efficacy. However, it's worth noting that our proposed F-Fidelity method still demonstrates consistent top-tier performance across all metrics and models, reinforcing its versatility and reliability even in scenarios where simpler methods perform well.

## 5.4 Practical Guidelines for F-Fidelity Usage

We provide in-depth analysis on the relation between model robustness and performance of F-Fidelity in Appendix D.2. The observed robustness patterns exhibit a direct correlation with F-Fidelity's effectiveness, which provides practical guidance. For tasks where models exhibit significant sensitivity to perturbations, such as computer vision and time series models, F-Fidelity should be the preferred choice over traditional fidelity metrics to ensure reliable explanation evaluation. In these domains, the substantial performance degradation under perturbation indicates vulnerability to OOD issues, making F-Fidelity's robustness-enhancing properties particularly valuable. However, for tasks where models demonstrate inherent robustness to feature removal, such as many NLP tasks, simpler fidelity metrics are preferred. This relationship between model robustness and evaluation method choice enables one to make informed decisions based on their specific domain characteristics and accuracy requirements. In Appendix D.2, we provide a simple robustness test by measuring model accuracy under different masking ratios to determine the most appropriate evaluation metric.

## 6 RELATED WORK

Existing methods for evaluating explanations can be generally divided into two categories according to whether ground truth explanations are available. Comparing to the ground truth is an intuitive way for explanation evaluation. For example, in time series data and graph data, there exist some synthetic datasets for explanation evaluation, including BA-Shapes (Ying et al., 2019), BA-Motifs (Luo et al., 2020), FreqShapes, SeqComb-UV, SeqComb-MV, and LowVar (Queen et al., 2024; Liu et al.). In computer Vision and Natural Language Processing, the important parts can also be obtained with human annotation. However, these methods suffer from the heavy labor of ground truth annotation.

The second category of evaluation methods assesses the explanation quality by comparing model outputs between the original input and inputs modified based on the generated explanations Zheng et al. (2024). For example, Class Activation Mapping (CAM) (Zhou et al., 2016) first compares the classification performance between the original and their GAP networks, which makes sure the explanation is faithful for the original network. Grad-CAM(Selvaraju et al., 2017) uses image occlusion to measure faithfulness. In the following work, Grad-CAM++ (Chattopadhay et al., 2018) uses three metrics to evaluate the performance of explanation methods, "Average Drop %", "% Increase in Confidence", and "Win %". In Adversarial Gradient Integration (Pan et al., 2021; Petsiuk et al., 2018), the authors use "Deletion Score" and "Insertion Score" to measure the faithfulness, where "Deletion Score" is to delete the attributions from the original input and "Insertion Score" is to insert attributions into one blank input according to the explanations. In (Samek et al., 2016), the LeRF/MoRF method is proposed to measure if the importance is consistent with the accuracy of the model. However, this group of metrics does not consider the effect of the OOD issues. In (Hooker et al., 2019), the author propose a new evaluation method ROAR to calculate the accuracy, which avoids the OOD problem by using retrain. ROAD (Rong et al., 2022) is introduced to solve information leakage and time-consuming issues. In the graph domain, GinX-Eval extends ROAR by introducing a fine-tuning strategy (Amara et al., 2023).

In the field of NLP, various methods are employed for evaluating faithfulness, as outlined in (Jacovi & Goldberg, 2020). These methods include axiomatic evaluation, predictive power evaluation, robustness evaluation, and perturbation-based evaluation, among others. Axiomatic evaluation involves testing explanations based on predefined principles (Jacovi & Goldberg, 2020; Adebayo et al., 2018; Liu et al., 2022; Wiegreffe et al., 2020). Predictive power evaluation operates on the premise that if explanations do not lead to the corresponding predictions, they are deemed unfaithful (Jacovi & Goldberg, 2020; Sia et al., 2023; Ye et al., 2021). Robustness evaluation examines whether explanations remain stable when there are minor changes in the input (Ju et al., 2021; Yin et al., 2021; Zheng et al., 2021), such as when input words with similar semantics produce similar outputs. Perturbation-based evaluation, one of the most widely used methods, assesses how explanations change when perturbed (Atanasova, 2024; Jain & Wallace, 2019). This approach is akin to MoRF and LeRF, where (DeYoung et al., 2019) measures prediction sufficiency and comprehensiveness by removing both unimportant and important features. For further information, please refer to the survey paper (Lyu et al., 2024).

## 7 CONCLUSION

In this paper, we introduced F-Fidelity, a robust framework for faithfulness evaluation in explainable AI. By leveraging a novel fine-tuning process, our method significantly mitigates the OOD problem that has plagued previous evaluation metrics. Through comprehensive experiments across multiple data modalities, we demonstrated that F-Fidelity consistently outperforms existing baselines in assessing the quality of explanations. Notably, our framework revealed a relationship between evaluation performance and ground truth explanation size under certain conditions, providing valuable insights into the nature of model explanations. In the future, we plan to explore alternative perturbation strategies, such as Gaussian blur. Additionally, we will investigate fine-tuning models with limited training data to further enhance the applicability and practicality of F-Fidelity, particularly in resource-constrained settings.

ACKNOWLEDGMENTS

This project was partially supported by NSF grants IIS-2331908 and CCF-2241057. The views and conclusions contained in this paper are those of the authors and should not be interpreted as representing any funding agencies.

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

## A   DETAILED ALGORITHM

In this section, we provided the detailed pipeline of our evaluation method F-Fidelity  as follows:

---

**Algorithm 1** Computing $FFid^+, FFid^-$

---

1: **Input:** model to be explained $f$, train set $\mathcal{D}_t$, explanation test set $\mathcal{D}_v$ with $n = |\mathcal{D}_v|$ samples, explainer $\psi$, explanation size $s$, original parameters $\alpha_{orig}^+$, $\alpha_{orig}^-$, fixed fraction $\beta$, training epochs $E$, sampling numbers $N$
2: **Output:** $FFid^+$, $FFid^-$
3: copy $f^r \leftarrow f$
4: **for** $e$ in range($E$) **do**
5:   **for** $(\mathbf{x}_i, y_i)$ in $\mathcal{D}_t$ **do**
6:     update $f^r$ by using $\mathcal{L}(f(\mathbf{x}_i - P_\beta \odot \mathbf{x}_i), y_i)$
7:   **end for**
8: **end for**
9: **for** $(\mathbf{x}_i, y_i)$ in $\mathcal{D}_v$ **do**
10:   $\mathbf{m} = \psi(\mathbf{x}_i)$        # obtain the explanation
11:   **for** $k$ in range($N$) **do**
12:     $\alpha^+ = \min(\alpha_{orig}^+, \frac{\beta td}{s})$
13:     $\alpha^- = \min(\alpha_{orig}^-, \frac{\beta td}{td-s})$
14:     $FFid^+[i, k] \leftarrow \mathbb{1}(y_i = f^r(\mathbf{x}_i)) - \mathbb{1}(y_i = f^r(\chi^+(\mathbf{x}_i, \alpha^+, s)))$
15:     $FFid^-[i, k] \leftarrow \mathbb{1}(y_i = f^r(\mathbf{x}_i)) - \mathbb{1}(y_i = f^r(\chi^-(\mathbf{x}_i, \alpha^-, s)))$
16:   **end for**
17:   $FFid^+[i] \leftarrow \frac{1}{N} \sum_{k=1}^N FFid^+[i, k]$
18:   $FFid^-[i] \leftarrow \frac{1}{N} \sum_{k=1}^N FFid^-[i, k]$
19: **end for**
20: $FFid^+ \leftarrow \frac{1}{|\mathcal{D}_v|} \sum_{(\mathbf{x}_i, y_i) \in \mathcal{D}_v} FFid^+[i]$
21: $FFid^- \leftarrow \frac{1}{|\mathcal{D}_v|} \sum_{(\mathbf{x}_i, y_i) \in \mathcal{D}_v} FFid^-[i]$
22: **Return** $FFid^+$, $FFid^-$

---

## B   PROOF OF THEOREM 1

We provide the proof by considering the following cases:
**Case 1:** $s \in [\max(\frac{\beta}{\alpha_{orig}^+} td, c_1), td]$
We have:

$$e(s) = \mathbb{E}_{\mathbf{X}, Y}(FFid^+(\psi, \alpha_{orig}^+, \beta, s))$$

$$= \mathbb{E}_{\mathbf{X}, Y}(\frac{1}{n} \sum_{(\mathbf{X}_i, Y_i) \in \mathcal{T}} \mathbb{1}(Y_i = f_r(\mathbf{X}_i)) - P(Y_i = f_r(\chi^+(\mathbf{X}_i, \alpha_{orig}^+, s)))),$$

where $f_r(\cdot)$ represent the finetuned model in Algorithm 1, which is assumed to be robust to up to $\beta td$ removals, i.e., $P(f_r(\mathbf{x} \odot \mathbf{m}) = Y) = P(Y|\mathbf{x} \odot \mathbf{m})$ if $\|\mathbf{m}\|_1 \geq td - \beta td$. Consequently,

$$e(s) = P_{\mathbf{X}, Y}(Y = f_r(\mathbf{X})) - P_{\mathbf{X}, Y}(Y = f_r(\chi^+(\mathbf{X}, \alpha_{orig}^+, s)))$$

$$= g(c_1, c_2, \cdots, c_r) - \sum_{j^r: j_i \leq c_i} P(J^r = j^r) g(c_1 - j_1, c_2 - j_2, \cdots, c_r - j_r)$$

$$= g(c_1, c_2, \cdots, c_r) - \mathbb{E}(g(c_1 - J_1, c_2 - J_2, \cdots, c_r - J_r)),$$

where $J^r$ is a multivariate hypergeometric vector with parameters $(n_1, n_2, \cdots, n_r, \beta td)$, where:

$$n_i = \begin{cases} c_i & \text{if } \sum_{i' \leq i} c_{i'} \leq s, \\ s - \sum_{i' < i} c_{i'} & \text{if } \sum_{i' < i} c_{i'} \leq s \leq \sum_{i' \leq i} c_{i'} . \\ 0 & \text{otherwise} \end{cases} \tag{7}$$

To explain the last equation, recall that $FFid^+$ first ranks the elements of the input based on Shap values, and chooses the top $s$ element. Then, it randomly and uniformly samples $\beta td$ elements from the top $s$ elements (as long as $\lfloor \alpha_{orig}^+ s \rfloor > \beta td$). So, if the $i$-th influence tier satisfies

$\sum_{i' \leq i} c_{i'} \leq s$, then all of its elements will be chosen among the top $s$ influential elements. If $\sum_{i' < i} c_{i'} \leq s \leq \sum_{i' \leq i} c_{i'}$, then $s - \sum_{i' < i} c_{i'}$ elements from the tier would be chosen among the top $s$, and otherwise, no elements will be chosen from the tier. So, in general, $n_i$ elements from each tier will be chosen, where $n_i$ is defined in equation 7. Next, $FFid$ samples $\beta td$ of these top $s$ elements and masks them. Thus, the number of elements masked in each tier follows a multivariate hypergeometric distribution, modeling the sampling of $\beta td$ elements from $n_1, n_2, \cdots, n_r$ elements of types $1, 2, \cdots, r$, respectively. As $s$ increases, the choice is *diluted* by the inclusion of the lowest tier elements in the top $s$ (since only $n_i$ for the tier with $\sum_{i' < i} c_{i'} \leq s \leq \sum_{i' \leq i} c_{i'}$ increases as $s$ is increased to $s + 1$). So, $J^r$ is decreasing (in terms of lexicographic ordering) as a function of $s$, and $c^r - J^r$ is increasing as a function of $s$. Since $e(s)$ is linearly related to $\mathbb{E}(g(c^r - J^r))$, it follows by the fact that $g(j_1, j_2, \cdots, j_r)$ is increasing with respect to the lexicographic ordering that $e(s)$ is decreasing in $s$.

**Case 2:** $s \in [0, c_1]$

The proof follows by similar arguments as in the previous case. We have:

$$e(s) = g(c_1, c_2, \cdots, c_r) - g(c_1 - \lfloor \alpha^+ s \rfloor, 0, 0, \cdots, 0),$$

It follows by the fact that $g(j_1, j_2, \cdots, j_r)$ is increasing with respect to the lexicographic ordering that $e(s)$ is increasing in $s$.

## C  DETAILED EXPERIMENT SETUP

In F-Fidelity, the hyperparameter $\beta$ is used in both fine-tuning and evaluation. We select 0.1 by default. For the sampling number, $N$, a larger number helps reduce the standard deviation but leads to increasing evaluation time. In this paper, we select a balance point of 50. The hyperparameter $\alpha^+$ and $\alpha^-$ are designed to alleviate the OOD problem, we choose $\alpha = \alpha^+ = \alpha^- = 0.5$. We provide hyperparameter sensitivity studies in Section D.9 and show comprehensive results in Section E.4. For the key parameter $\beta$, we further provide an in-depth analysis in Section D.3. For baselines, we use their default values as suggested in their original papers or codes.

We explore two architectures, ResNet and ViT for the image classification tasks. We use ResNet-9 [3] and a 6-layer ViT as backbones for both  CIFAR-100  and  Tiny-Imagenet . In the ResNet architecture, the hidden dimensions are set to $[64, 128, 128, 256, 512, 512, 512, 512]$. For the ViT model, we use a patch size of 4, with a patch embedding dimension of 128. The backbone consists of 6 attention layers, each with 8 heads for multi-head attention. The final output hidden dimension of the ViT encoder is 256. In the training stage, we set the learning rate and weight decay to 1E-4 for ResNet and set the learning rate to 1E-3 and weight decay to 1E-4 for ViT. We use Adam as the optimizer and the training epochs are 100 for ResNet and 200 for ViT. During fine-tuning, we use the same hyperparameters.

For the time series classification task, we follow the TimeX (Queen et al., 2023) to use a simple LSTM for both Boiler and PAM datasets. The simple LSTM model contains 3 bidirectional LSTM layers with 128 hidden embedding sizes. We use AdamW as the optimizer with a learning rate of 1E-3 and weight decay is 1E-2.

For the natural language task, we use LSTM (Hochreiter & Schmidhuber, 1997) and Transformer as the backbone. the LSTM has one hidden layer with the dimension set to 128. In Transformer, the hidden dimension is set to 512. The number of Transformer layers for SST2 and BoolQ are 2 and 4. The head of the Transformer layers for SST2 and BoolQ are 4 and 8. For all datasets and architectures, we use Adam optimizer (Kingma & Ba, 2015) with default learning rate 1E-4, training epochs 100.

## D  EXTRA EXPERIMENTAL STUDY

### D.1  EXPERIMENTS WITH OTHER BACKBONES.

**ViT as Backbone for Image Classification.**    To further validate the robustness of our F-Fidelity framework across different model architectures, we conducted additional experiments using ViT (Dosovitskiy et al., 2020) as the backbone for image classification on both CIFAR-100 and

---

[3]https://jovian.com/tessdja/resnet-practice-cifar100-resnet

TinyImageNet datasets. We utilize the SG-SQ (Smilkov et al., 2017) to generate explanations. For the settings, we follow our main experimental setup. As shown in Table 5, F-Fidelity consistently achieves best correlations across all metrics for both datasets, outperforming other methods. This strong performance on ViT models further emphasizes the versatility and effectiveness of F-Fidelity in evaluating explanations across different deep learning architectures.

Table 5: Spearman rank correlation results on CIFAR-100 and Tiny-Imagenet dataset with ViT.

| | Correlation | CIFAR-100 | | | | Tiny-Imagenet | | | |
|---|---|---|---|---|---|---|---|---|---|
| | | Fidelity | ROAR | R-Fidelity | F-Fidelity | Fidelity | ROAR | R-Fidelity | F-Fidelity |
| Macro Corr. | MoRF vs GT ↓ | $-0.94_{\pm0.00}$ | $-0.94_{\pm0.00}$ | $\mathbf{-1.00}_{\pm0.00}$ | $\mathbf{-1.00}_{\pm0.00}$ | $-0.82_{\pm0.02}$ | $-0.83_{\pm0.00}$ | $\mathbf{-1.00}_{\pm0.00}$ | $\mathbf{-1.00}_{\pm0.00}$ |
| | LeRF vs GT ↑ | $\mathbf{1.00}_{\pm0.00}$ | $\mathbf{1.00}_{\pm0.00}$ | $\mathbf{1.00}_{\pm0.00}$ | $\mathbf{1.00}_{\pm0.00}$ | $\mathbf{1.00}_{\pm0.00}$ | $\mathbf{1.00}_{\pm0.00}$ | $\mathbf{1.00}_{\pm0.00}$ | $\mathbf{1.00}_{\pm0.00}$ |
| | MoRF vs LeRF ↓ | $-0.94_{\pm0.08}$ | $-0.94_{\pm0.00}$ | $\mathbf{-1.00}_{\pm0.00}$ | $\mathbf{-1.00}_{\pm0.00}$ | $-0.82_{\pm0.02}$ | $-0.83_{\pm0.00}$ | $\mathbf{-1.00}_{\pm0.00}$ | $\mathbf{-1.00}_{\pm0.00}$ |
| Micro Corr. | MoRF vs GT ↓ | $-0.68_{\pm0.40}$ | $-0.91_{\pm0.07}$ | $\mathbf{-1.00}_{\pm0.01}$ | $\mathbf{-1.00}_{\pm0.00}$ | $-0.74_{\pm0.27}$ | $-0.76_{\pm0.10}$ | $\mathbf{-1.00}_{\pm0.00}$ | $\mathbf{-1.00}_{\pm0.00}$ |
| | LeRF vs GT ↑ | $\mathbf{1.00}_{\pm0.01}$ | $0.94_{\pm0.21}$ | $\mathbf{1.00}_{\pm0.00}$ | $\mathbf{1.00}_{\pm0.00}$ | $\mathbf{1.00}_{\pm0.00}$ | $\mathbf{1.00}_{\pm0.00}$ | $\mathbf{1.00}_{\pm0.00}$ | $\mathbf{1.00}_{\pm0.00}$ |
| | MoRF vs LeRF ↓ | $-0.69_{\pm0.39}$ | $-0.77_{\pm0.44}$ | $\mathbf{-1.00}_{\pm0.01}$ | $\mathbf{-1.00}_{\pm0.00}$ | $-0.74_{\pm0.27}$ | $-0.76_{\pm0.10}$ | $\mathbf{-1.00}_{\pm0.00}$ | $\mathbf{-1.00}_{\pm0.00}$ |
| Micro Rank | MoRF vs GT ↓ | $3.47_{\pm0.94}$ | $3.16_{\pm0.67}$ | $1.05_{\pm0.22}$ | $\mathbf{1.00}_{\pm0.00}$ | $3.26_{\pm0.44}$ | $3.63_{\pm0.48}$ | $\mathbf{1.00}_{\pm0.00}$ | $\mathbf{1.00}_{\pm0.00}$ |
| | LeRF vs GT ↓ | $1.16_{\pm0.36}$ | $\mathbf{1.11}_{\pm0.31}$ | $1.32_{\pm0.57}$ | $1.32_{\pm0.57}$ | $\mathbf{1.00}_{\pm0.00}$ | $\mathbf{1.00}_{\pm0.00}$ | $\mathbf{1.00}_{\pm0.00}$ | $\mathbf{1.00}_{\pm0.00}$ |
| | MoRF vs LeRF ↓ | $3.58_{\pm0.75}$ | $3.16_{\pm0.67}$ | $1.05_{\pm0.22}$ | $\mathbf{1.00}_{\pm0.00}$ | $3.26_{\pm0.44}$ | $3.63_{\pm0.48}$ | $\mathbf{1.00}_{\pm0.00}$ | $\mathbf{1.00}_{\pm0.00}$ |

**CNN as Backbone for Time Series Classification.** To verify the effectiveness of F-Fidelity on Time Series, we conduct experiments with CNN as the classifier by following the setting in Section 5.2. As Table 6 shows, F-Fidelity consistently outperforms baselines across all metrics including macro and micro correlations. Similarly, we observe the same phenomenon as shown in Table 3, the Fidelity and ROAR suffer OOD problem. Under some sparsity, the correlation is not available for the same rank while F-Fidelity and R-Fidelity perform well. These results on CNN models highlight the versatility and effectiveness of F-Fidelity in assessing explanations across various deep learning architectures.

Table 6: Spearman ranks correlations and ranks on time series datasets with CNN as the classifier. The best performance is marked as bold. The "-" means the correlation can't be obtained because the different explanations have the same rank.

| | Correlation | PAM | | | | Boiler | | | |
|---|---|---|---|---|---|---|---|---|---|
| | | Fidelity | ROAR | R-Fidelity | F-Fidelity | Fidelity | ROAR | R-Fidelity | F-Fidelity |
| Macro Corr. | MoRF vs GT ↓ | $0.78_{\pm0.10}$ | $-0.90_{\pm0.06}$ | $\mathbf{-1.00}_{\pm0.00}$ | $\mathbf{-1.00}_{\pm0.00}$ | $\mathbf{-1.00}_{\pm0.00}$ | $\mathbf{-1.00}_{\pm0.00}$ | $\mathbf{-1.00}_{\pm0.00}$ | $\mathbf{-1.00}_{\pm0.00}$ |
| | LeRF vs GT ↑ | $0.96_{\pm0.14}$ | $-0.71_{\pm0.10}$ | $\mathbf{1.00}_{\pm0.00}$ | $\mathbf{1.00}_{\pm0.00}$ | $0.91_{\pm0.07}$ | $0.86_{\pm0.07}$ | $\mathbf{1.00}_{\pm0.00}$ | $\mathbf{1.00}_{\pm0.00}$ |
| | MoRF vs LeRF ↓ | $0.68_{\pm0.18}$ | $\mathbf{0.61}_{\pm0.11}$ | $\mathbf{-1.00}_{\pm0.00}$ | $\mathbf{-1.00}_{\pm0.00}$ | $-0.91_{\pm0.07}$ | $-0.86_{\pm0.07}$ | $\mathbf{-1.00}_{\pm0.00}$ | $\mathbf{-1.00}_{\pm0.00}$ |
| Micro Corr. | MoRF vs GT ↓ | $-0.02_{\pm0.62}$ | $-0.39_{\pm0.66}$ | $\mathbf{-1.00}_{\pm0.00}$ | $\mathbf{-1.00}_{\pm0.00}$ | - | - | $\mathbf{-0.94}_{\pm0.13}$ | $-0.92_{\pm0.18}$ |
| | LeRF vs GT ↑ | $0.93_{\pm0.15}$ | $0.68_{\pm0.53}$ | $\mathbf{1.00}_{\pm0.01}$ | $\mathbf{1.00}_{\pm0.00}$ | - | - | $\mathbf{0.93}_{\pm0.11}$ | $\mathbf{0.93}_{\pm0.11}$ |
| | MoRF vs LeRF ↓ | $-0.03_{\pm0.58}$ | $-0.12_{\pm0.61}$ | $\mathbf{-1.00}_{\pm0.00}$ | $\mathbf{-1.00}_{\pm0.00}$ | - | - | $-0.93_{\pm0.12}$ | $\mathbf{-0.94}_{\pm0.11}$ |
| Micro Rank | MoRF vs GT ↓ | $3.00_{\pm1.26}$ | $2.00_{\pm1.08}$ | $\mathbf{1.00}_{\pm0.00}$ | $1.11_{\pm0.45}$ | $3.16_{\pm0.87}$ | $2.63_{\pm0.87}$ | $\mathbf{1.11}_{\pm0.31}$ | $1.32_{\pm0.46}$ |
| | LeRF vs GT ↓ | $2.68_{\pm1.22}$ | $2.84_{\pm1.42}$ | $\mathbf{1.00}_{\pm0.00}$ | $\mathbf{1.00}_{\pm0.00}$ | $3.11_{\pm0.64}$ | $3.53_{\pm0.50}$ | $1.47_{\pm0.60}$ | $\mathbf{1.21}_{\pm0.41}$ |
| | MoRF vs LeRF ↓ | $3.68_{\pm0.46}$ | $2.79_{\pm1.15}$ | $\mathbf{1.00}_{\pm0.00}$ | $1.11_{\pm0.45}$ | $3.21_{\pm0.41}$ | $3.05_{\pm0.22}$ | $1.32_{\pm0.46}$ | $\mathbf{1.16}_{\pm0.36}$ |

**Transformer as Backbone for Natural Language Classification.** To further validate our findings with modern architectures, we conducte experiments using Transformer models on both SST2 and BoolQ datasets. As shown in Table 7, the results demonstrate interesting patterns. On SST2, while all methods perform well, F-Fidelity shows slight advantages, particularly in micro correlations and rankings. For BoolQ, all methods achieve nearly perfect correlations, suggesting that Transformer models on this dataset exhibit strong robustness to perturbations, making the differences between evaluation methods less pronounced. These results align with our earlier observations about the inherent robustness of NLP models, while still demonstrating the reliability and slight advantages of F-Fidelity in more challenging scenarios.

## D.2 RELATIONSHIP BETWEEN MODEL ROBUSTNESS WITH F-FIDELITY PERFORMANCE

To understand how model robustness influences F-Fidelity's effectiveness, we conduct a systematic evaluation across three domains: computer vision (CIFAR-100), time series (Boiler), and NLP (SST2). For each domain, we test multiple architectures - ResNet and Transformer for CIFAR-100, LSTM and CNN for Boiler, and LSTM and Transformer for SST2. We measure model robust-

Table 7: Spearman rank correlation and rank results with Transformer model on SST2 and BoolQ datasets.

| Correlation | | SST2 | | | BoolQ | | |
|---|---|---|---|---|---|---|---|
| | | Fidelity | R-Fidelity | F-Fidelity | Fidelity | R-Fidelity | F-Fidelity |
| Macro Corr. | MoRF vs GT ↓ | $-0.94_{\pm0.00}$ | $\mathbf{-1.00}_{\pm0.00}$ | $\mathbf{-1.00}_{\pm0.00}$ | $\mathbf{-1.00}_{\pm0.00}$ | $\mathbf{-1.00}_{\pm0.00}$ | $\mathbf{-1.00}_{\pm0.00}$ |
| | LeRF vs GT ↑ | $\mathbf{1.00}_{\pm0.00}$ | $\mathbf{1.00}_{\pm0.00}$ | $\mathbf{1.00}_{\pm0.00}$ | $\mathbf{1.00}_{\pm0.00}$ | $\mathbf{1.00}_{\pm0.00}$ | $\mathbf{1.00}_{\pm0.00}$ |
| | MoRF vs LeRF ↓ | $-0.94_{\pm0.00}$ | $\mathbf{-1.00}_{\pm0.00}$ | $\mathbf{-1.00}_{\pm0.00}$ | $\mathbf{-1.00}_{\pm0.00}$ | $\mathbf{-1.00}_{\pm0.00}$ | $\mathbf{-1.00}_{\pm0.00}$ |
| Micro Corr. | MoRF vs GT ↓ | $-0.93_{\pm0.07}$ | $-0.97_{\pm0.05}$ | $\mathbf{-1.00}_{\pm0.01}$ | $\mathbf{-1.00}_{\pm0.01}$ | $\mathbf{-1.00}_{\pm0.01}$ | $\mathbf{-1.00}_{\pm0.00}$ |
| | LeRF vs GT ↑ | $0.98_{\pm0.04}$ | $\mathbf{0.99}_{\pm0.02}$ | $0.99_{\pm0.03}$ | $\mathbf{1.00}_{\pm0.00}$ | $\mathbf{1.00}_{\pm0.00}$ | $\mathbf{1.00}_{\pm0.00}$ |
| | MoRF vs LeRF ↓ | $-0.92_{\pm0.09}$ | $-0.97_{\pm0.05}$ | $\mathbf{-0.99}_{\pm0.02}$ | $\mathbf{-1.00}_{\pm0.00}$ | $\mathbf{-1.00}_{\pm0.00}$ | $\mathbf{-1.00}_{\pm0.00}$ |
| Micro Rank | MoRF vs GT ↓ | $2.05_{\pm0.89}$ | $1.32_{\pm0.46}$ | $\mathbf{1.05}_{\pm0.22}$ | $\mathbf{1.00}_{\pm0.00}$ | $\mathbf{1.00}_{\pm0.00}$ | $\mathbf{1.00}_{\pm0.00}$ |
| | LeRF vs GT ↓ | $\mathbf{1.16}_{\pm0.49}$ | $1.26_{\pm0.55}$ | $\mathbf{1.16}_{\pm0.36}$ | $\mathbf{1.00}_{\pm0.00}$ | $\mathbf{1.00}_{\pm0.00}$ | $\mathbf{1.00}_{\pm0.00}$ |
| | MoRF vs LeRF ↓ | $2.05_{\pm0.97}$ | $1.32_{\pm0.46}$ | $\mathbf{1.11}_{\pm0.45}$ | $\mathbf{1.00}_{\pm0.00}$ | $\mathbf{1.00}_{\pm0.00}$ | $\mathbf{1.00}_{\pm0.00}$ |

Table 8: Classification accuracy (%) comparison between original and fine-tuned models under different perturbation ratios.

| Dataset | Architecture | Model | 0% | 5% | 10% |
|---|---|---|---|---|---|
| CIFAR-100 | ResNet | Original | 73.23 | 56.70 | 41.81 |
| | | Fine-tuned | 73.10 | 73.12 | 72.97 |
| | ViT | Original | 50.25 | 46.19 | 39.87 |
| | | Fine-tuned | 46.91 | 47.05 | 47.30 |
| Boiler | LSTM | Original | 80.14 | 67.78 | 61.53 |
| | | Fine-tuned | 77.22 | 72.64 | 73.19 |
| | CNN | Original | 82.22 | 68.89 | 57.92 |
| | | Fine-tuned | 86.39 | 74.03 | 63.89 |
| SST2 | LSTM | Original | 82.45 | 80.28 | 79.82 |
| | | Fine-tuned | 82.68 | 80.16 | 79.24 |
| | Transformer | Original | 80.85 | 77.29 | 73.28 |
| | | Fine-tuned | 80.73 | 81.08 | 78.90 |

ness through classification accuracy under different perturbation levels (0%, 5%, and 10%), where perturbations are implemented through domain-specific masking operations: zeroing pixels for images, masking feature values for time series, and replacing token with zeros for text. For each model-architecture pair, we compare the performance between the original model and its fine-tuned version under these perturbation conditions.

Table 8 reveals distinct robustness patterns across different domains and architectures. In computer vision, models exhibit high sensitivity to perturbations - the original ResNet on CIFAR-100 shows dramatic accuracy degradation under even small perturbations, while its fine-tuned counterpart maintains stable performance. Time series models demonstrate moderate sensitivity, with original models showing notable but less severe accuracy drops under perturbations, which fine-tuning helps to stabilize. In contrast, NLP models display inherent robustness, maintaining relatively stable performance even under perturbations regardless of fine-tuning.

These robustness patterns directly correlate with F-Fidelity's effectiveness. The performance improvements provided by F-Fidelity are most pronounced in domains where models show high sensitivity to perturbations (computer vision and time series). This is because traditional fidelity metrics in these domains suffer from unreliable evaluations due to OOD inputs, which our fine-tuning strategy effectively mitigates. However, in NLP tasks where models possess natural robustness to perturbations, the OOD problem is less severe, consequently diminishing the relative advantages of F-Fidelity. This relationship provides practical guidance for choosing evaluation metrics - F-Fidelity should be preferred in domains where models show high sensitivity to perturbations, while simpler metrics may suffice for naturally robust domains.

### D.3 ANALYSIS OF DECISION BOUNDARY PRESERVATION UNDER RANDOM MASKING FINE-TUNING

This section addresses whether fine-tuning the classification model with random masking (up to fraction $\beta$ of input elements) significantly changes the model's decision boundaries. We examine this by looking at both global classification patterns and local decision characteristics. We consider two aspects of the model's decision boundaries as follows.

- **Global Decision Boundary**: The global decision boundary refers to the overall separation between different classes in the input space. It captures the main structure of the data distribution and represents the classifier's ability to distinguish between classes on a broad scale.
- **Local Decision Boundary**: The local decision boundary characterizes how the model responds to small perturbations around individual data points.

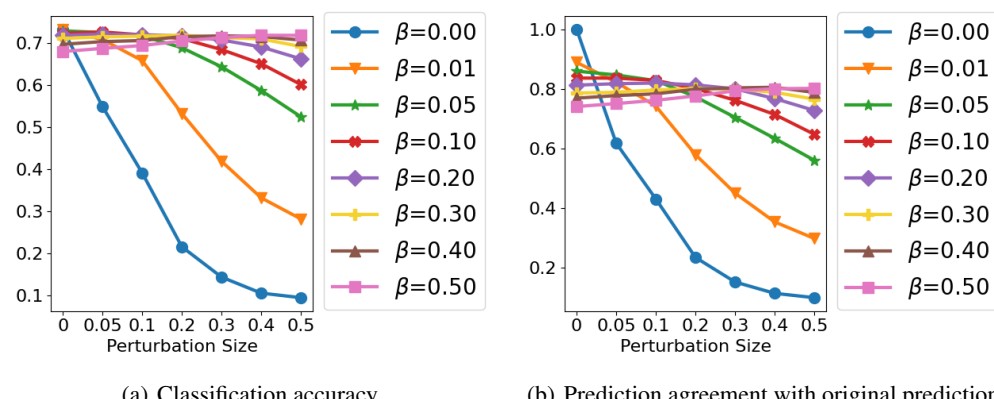

(a) Classification accuracy         (b) Prediction agreement with original prediction

Figure 1: Effects of random masking fine-tuning on model behavior. Higher fine-tuning $\beta$ values improve robustness to perturbations but reduce agreement with the original model's predictions.

Our random masking fine-tuning aims to maintain the model's global classification behavior while improving its stability to small input changes. The random masking process applies uniform noise across all inputs, regardless of their class. This helps the model learn to handle missing information while keeping its main classification patterns intact. Random masking fine-tuning maintains model performance through several key principles. First, the random masking operation applies a uniform noise distribution across the input space. Since this noise is class-agnostic and uniformly distributed, it preserves the statistical properties of the underlying class distributions. Second, random masking acts as a form of dropout regularization (Srivastava et al., 2014), which has been shown to improve model generalization while maintaining core decision boundaries. Third, the introduction of masked inputs during training encourages the model to develop smoother decision boundaries in local neighborhoods, as demonstrated by Bishop (1995) in the context of noise injection.

To empirically evaluate random masking fine-tuning, we conduct experiments with ResNet on CIFAR-100. We examine both global decision boundary preservation and local boundary characteristics through systematic perturbation analysis. For global boundary preservation, we compare classification accuracy between the original and fine-tuned models on clean test data, using 0% masking as the baseline reference point. We also measure prediction agreement between these models to assess consistency in their decision-making. To analyze local boundary characteristics, we apply incremental perturbations during testing by randomly masking input elements, with perturbation sizes ranging from 0.05 to 0.50. At each perturbation level, we track both classification accuracy and prediction agreement with the original model. Classification accuracy reveals the model's robustness to perturbations, while prediction agreement quantifies the consistency of decision boundaries under noise. The masking ratio $\beta$ during fine-tuning controls the balance between preserving global boundaries and smoothing local ones.

Our experimental results demonstrate that random masking fine-tuning effectively balances global decision boundary preservation with local boundary smoothing. The fine-tuned models maintain

comparable baseline accuracy to the original model while showing significantly improved robustness to perturbations. On clean test data, the accuracy difference between original and fine-tuned models remains small, indicating preserved global classification behavior. The high prediction agreement between original and fine-tuned models on clean data further confirms the preservation of global decision patterns. Meanwhile, the smoother local decision boundaries manifest through sustained performance under perturbations, with fine-tuned models showing notably accuracy change compared to the original model.

The masking ratio $\beta$ plays a crucial role in model behavior under perturbations, as evidenced in Figure 1. In terms of accuracy (Figure 1(a)), models fine-tuned with larger $\beta$ values demonstrate remarkable resilience, maintaining accuracy above 60% even under 50% input perturbation, while the original model ($\beta = 0$) drops below 40% accuracy with just 10% perturbation. The prediction agreement results (Figure 1(b)) show a trade-off. Higher $\beta$ values lead to better accuracy under perturbations and result in lower agreement with the original model's predictions. This suggests that stronger fine-tuning with larger $\beta$ values causes the model to learn more robust but slightly different decision boundaries compared to the original model. Models with moderate $\beta$ values (like 0.10) maintain a better balance between perturbation resistance and consistency with the original model's behavior.

### D.4 PRESERVATION OF EXPLANATION AFTER FINE-TUNING

To further validate that our fine-tuning strategy preserves meaningful explanations, we conduct a qualitative analysis on Tiny-Imagenet. We visualize and compare the attribution maps generated for both the original and fine-tuned models using GradCAM. As shown in Table 9, the attribution maps from both models highlight similar regions of importance, with only minor variations in the highlight intensity. This visual consistency suggests that while our fine-tuning process improves model robustness to perturbations, it maintains the model's original attention to meaningful features, thereby preserving the validity of the generated explanations.

Table 9: The explanation visualization comparison between original and fine-tuned model.

### D.5 Evaluation with LayerCAM as Alternative Explanation Method

To further validate the robustness of our method across different explanation techniques, we conduct additional experiments using LayerCAM (Jiang et al., 2021), an alternative explanation method to GradCAM. Following the same experimental setup as in Table 1, we evaluate the performance using ResNet architecture on the CIFAR-100 dataset. As shown in Table 10, F-Fidelity maintains its superior performance with LayerCAM, exhibiting consistent patterns with our GradCAM results. This consistency across different explanation methods reinforces the generalizability of our evaluation framework and its ability to assess various types of explanation techniques reliably.

Table 10: Spearman rank correlation and rank results on CIFAR-100 dataset with ResNet explained by LayerCAM.

| | Correlation | Fidelity | ROAR | R-Fidelity | F-Fidelity |
|---|---|---|---|---|---|
| Macro Corr. | MoRF vs GT ↓ | $0.77_{\pm 0.06}$ | $0.98_{\pm 0.03}$ | $0.26_{\pm 0.00}$ | $\mathbf{-0.26}_{\pm 0.00}$ |
| | LeRF vs GT ↑ | $\mathbf{1.00}_{\pm 0.00}$ | $\mathbf{1.00}_{\pm 0.00}$ | $\mathbf{1.00}_{\pm 0.00}$ | $\mathbf{1.00}_{\pm 0.00}$ |
| | MoRF vs LeRF ↓ | $0.77_{\pm 0.06}$ | $0.98_{\pm 0.03}$ | $0.26_{\pm 0.00}$ | $\mathbf{-0.26}_{\pm 0.00}$ |
| Micro Corr. | MoRF vs GT ↓ | $0.53_{\pm 0.40}$ | $0.81_{\pm 0.20}$ | $-0.02_{\pm 0.22}$ | $\mathbf{-0.50}_{\pm 0.61}$ |
| | LeRF vs GT ↑ | $0.99_{\pm 0.02}$ | $0.99_{\pm 0.04}$ | $\mathbf{1.00}_{\pm 0.00}$ | $\mathbf{1.00}_{\pm 0.00}$ |
| | MoRF vs LeRF ↓ | $0.53_{\pm 0.41}$ | $0.81_{\pm 0.20}$ | $-0.02_{\pm 0.22}$ | $\mathbf{-0.50}_{\pm 0.61}$ |
| Micro Rank | MoRF vs GT ↓ | $2.95_{\pm 0.23}$ | $3.95_{\pm 0.23}$ | $1.89_{\pm 0.66}$ | $\mathbf{1.21}_{\pm 0.42}$ |
| | LeRF vs GT ↓ | $2.53_{\pm 1.39}$ | $1.95_{\pm 1.18}$ | $1.05_{\pm 0.23}$ | $\mathbf{1.00}_{\pm 0.00}$ |
| | MoRF vs LeRF ↓ | $2.89_{\pm 0.32}$ | $3.95_{\pm 0.23}$ | $1.89_{\pm 0.56}$ | $\mathbf{1.21}_{\pm 0.42}$ |

### D.6 Case Study

To provide the intuitional verification of our method, we conduct a case study with an image from the Tiny-Imagenet dataset. We first use GradCAM to obtain the explanation. To get various degraded explanations, we follow the previous setting to add noise, which is a list of $[0.2, 0.4, 0.6, 0.8, 1.0]$, to the explanation to obtain the ground truth ranking of explanations. We compare baselines and F-Fidelity and show the results in Figure 2. We observe that the Fidelity and ROAR methods fail to differentiate explanations with noise levels of 0.2 and 0.4 when the sparsity is below certain thresholds (e.g., 25%). However, the R-Fidelity and F-Fidelity have a good performance in such cases and F-Fidelity has a better performance than R-Fidelity from the micro correlation results.

### D.7 Ablation Study

To evaluate the effectiveness of our proposed F-Fidelity framework, we conduct ablation studies comparing three evaluation setups: the original Fidelity metric, Fidelity with fine-tuning (Fidelity+Fine-tune), and F-Fidelity. We utilize the CIFAR-100 dataset for these experiments, employing a ResNet architecture as the backbone for the classifier. The explainers are based on SG-SQ.

As shown in Figure 3, across all three correlation metrics—"MoRF vs GT", "LeRF vs GT", and "MoRF vs LeRF"—F-Fidelity consistently outperforms the other methods. Notably, Fidelity+Fine-tune shows a marked improvement over the original Fidelity, indicating that the fine-tuning step enhances the robustness of the evaluation. F-Fidelity further advances this by effectively addressing the OOD issue, resulting in the most faithful and reliable rankings of the explainers.

### D.8 Empirical Verification of determining the Explanation Size

In this section, we demonstrate empirically that the $FFid$ metric can be used to deduce the size or sparsity of explanations, complementing the theoretical analysis of Section 4. Specifically, we consider the colored-MNIST dataset (Arjovsky et al., 2019), where the explanation corresponds to the pixels representing the digit in each image. To control the explanation size, we rescale and crop the image samples so that the digit pixels occupy $\gamma \in \{0.1, 0.15, 0.2, 0.25\}$ fraction of the total image area. Our goal is to show empirically that the $FFid$ metric can recover the value of $\gamma$, thus revealing the explanation size. A three-convolution-layer model is used as the pre-trained model.

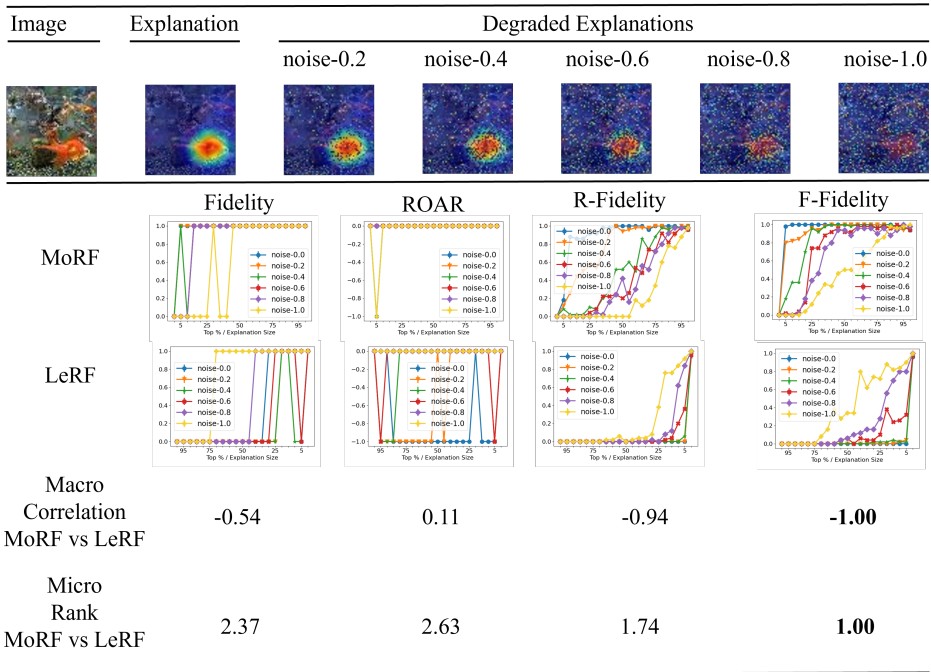

Figure 2: Case study of a sample in Tiny-Imagenet. We use GradCAM to obtain the explanation and get degraded explanations by adding random noise to the explanation. The noise level is a list of $[0.2, 0.4, 0.6, 0.8, 1.0]$. We visualize the Fidelity score and Correlation on different sparsity.

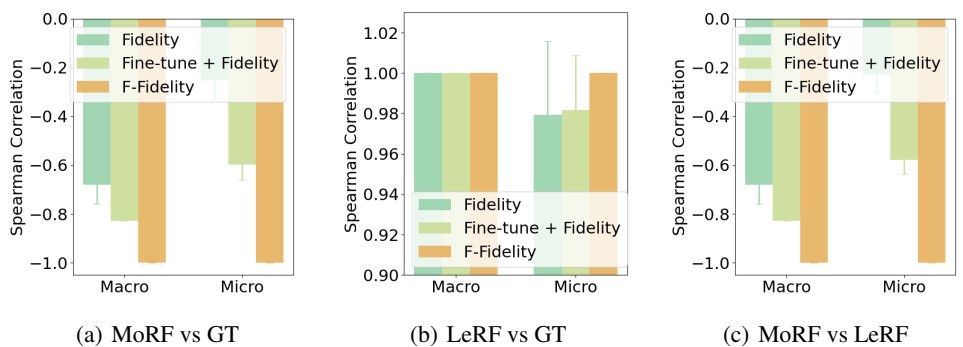

| (a) MoRF vs GT | (b) LeRF vs GT | (c) MoRF vs LeRF |
|---|---|---|

Figure 3: The ablation study between ground truth explanation size $\gamma$ and $FFid^+$ in F-Fidelity.

Following the terminology of Section 4, the pixels can be divided into two tiers: those belonging to the digit (forming the explanation) and those in the background. The explanation size is thus $c_1 = \gamma td$. According to Theorem 1, we expect $FFid^+$ to increase for $s < c_1$ and decrease for $s > \max\left(\frac{\beta}{\alpha^+}td, \gamma td\right)$.

To verify this behavior, we evaluate $FFid^+$ across three settings of $\alpha^+$ and five settings of $\beta$. As shown in Figure 4, the ground truth explanation size can be identified from the flat area in $RFid^+$ curves. Two critical points emerge on the x-axis: the explanation size $\gamma$ and the point where the removal attributions equal the explanation size ($\beta/\alpha^+$). The region between these points forms a plateau that enables reliable comparison of explanations. When $\frac{\beta}{\alpha^+}td < c_1$, the point where $FFid^+$ changes direction corresponds to the explanation size $\gamma$, allowing us to recover the true explanation size by evaluating $FFid^+$ over different $\beta$ values.

While our theoretical analysis assumes uniform importance weights across explanation components, real-world applications show some deviation from this idealized behavior, manifesting as smooth transitions rather than sharp changes in the $FFid^+$ curves. Nevertheless, the empirical results validate the theoretical predictions and demonstrate the utility of $FFid$ metrics in determining explanation sizes.

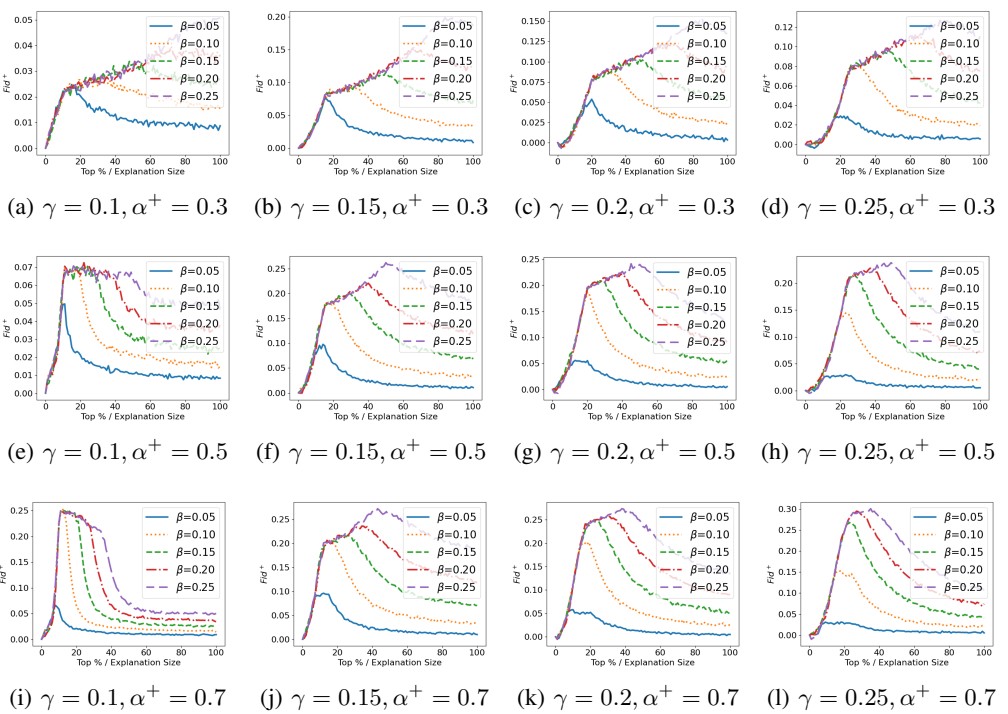

Figure 4: The relationship between ground truth explanation size $\gamma$ and $RFid^+$ in F-Fidelity. We compare different $\alpha^+$ and $\beta$ in the evaluation stage. We observe that the ground truth can be inferred from the $RFid^+$.

### D.9 HYPERPARAMETER SENSITIVITY STUDY

In this part, we assess the robustness of F-Fidelity to hyperparameter choices. We select a subset of 400 samples with an explanation size ratio of 0.2 from the colored-MNIST dataset (Arjovsky et al., 2019). we have three key hyperparameters: the number of sampling iterations $N$, the ratio $\alpha^+/\alpha^-$, and $\beta$. For each hyperparameter, we explored a range of values to understand their impact on the performance of F-Fidelity. We evaluated the method's performance using three Macro Spearman correlations, with additional detailed results on $RFid^+$, $RFid^-$, and Micro Spearman correlations provided in Appendix E.4.

Figure 5 presents the results of our hyperparameter sensitivity study. The findings demonstrate that F-Fidelity exhibits remarkable stability across a wide range of hyperparameter settings. This consistency is observed across all three Macro Spearman correlations, indicating that the performance of F-Fidelity is not overly sensitive to specific hyperparameter choices.

### D.10 COMPARING DIFFERENT EXPLANATION METHODS

To demonstrate F-Fidelity's ability to differentiate between explanation methods of varying quality, we conduct a comparative study between GradCAM and Saliency Map (Simonyan et al., 2013) on both Tiny-Imagenet and ImageNet. GradCAM typically produces more focused explanations by utilizing class-specific gradient information at higher-level feature maps, while Saliency Map operates directly on pixel-level gradients.

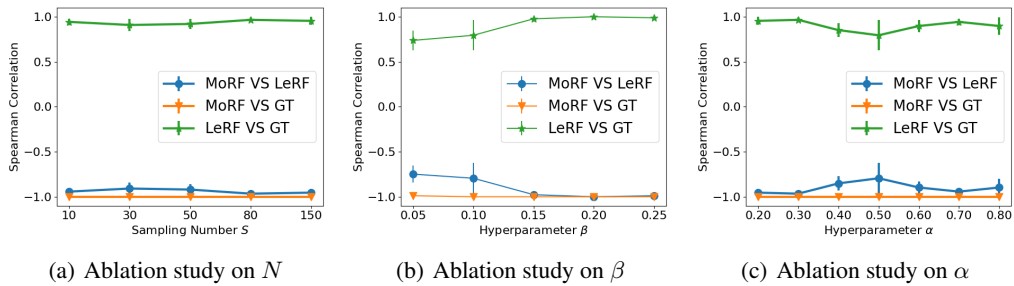

| (a) Ablation study on $N$ | (b) Ablation study on $\beta$ | (c) Ablation study on $\alpha$ |

Figure 5: Ablation study on sampling number $N$, $\beta$, and $\alpha = \alpha^+ = \alpha^-$. We report the Macro Spearman rank correlations.

Our evaluation on Tiny-Imagenet confirms this expected difference. Across our test set, GradCAM achieves better performance with higher $FFid^+$ (0.2286 vs 0.2020) and lower $FFid^-$ (0.0547 vs 0.1050) scores than Saliency Map. Note that both explanation methods are used to generate explanations with respect to the ground truth label. Thus, the FFid values are negative if the original prediction is incorrect.

Table 11: Comparative evaluation of GradCAM and Saliency Map explanations using F-Fidelity on Tiny-Imagenet. We show five cases with their original images and corresponding explanation visualizations. The $FFid^+$ and $FFid^-$ scores are averaged over different sparsity levels, with bold scores indicating better performance. The $FFid^+$ and $FFid^-$ scores of case 5 are negative because the original image is misclassified. The bottom row reports metrics averaged across all samples in the test set.

| Image | Explanation Visualization | | | Evaluation Metrics | | |
|---|---|---|---|---|---|---|
| | Original | GradCAM | Saliency Map | Method | $FFid^+ \uparrow$ | $FFid^- \downarrow$ |
| Case 1 | | | | GradCAM | **0.8743** | **0.1267** |
| | | | | Saliency Map | 0.8638 | 0.3342 |
| Case 2 | | | | GradCAM | **0.3838** | 0.029 |
| | | | | Saliency Map | 0.3143 | **0.0048** |
| Case 3 | | | | GradCAM | **0.2095** | **0.0190** |
| | | | | Saliency Map | 0.0752 | 0.0314 |
| Case 4 | | | | GradCAM | **0.4552** | **0.0486** |
| | | | | Saliency Map | 0.1095 | 0.1000 |
| Case 5 | | | | GradCAM | **-0.0057** | **-0.0048** |
| | | | | Saliency Map | -0.0086 | -0.0029 |
| Average over All Images in The Test Set | | | | GradCAM | **0.2286** | **0.0547** |
| | | | | Saliency Map | 0.2020 | 0.1050 |

To verify our method's effectiveness on high-dimensional data, we further evaluate on a subset of ImageNet (5000 images, 224×224×3 resolution) using ResNet-18 as the backbone. Unlike our previous experiments where we fine-tuned on training data and evaluated explainers on a separate test set, here we fine-tune and evaluate on the same image set to enable controlled comparison between explainers. Note that this fine-tuned model is specifically optimized for this dataset and should only be used to compare explainer performance within this set. Following the same protocol as our Tiny-Imagenet experiments, we fine-tune the model using Adam optimizer with default settings and

evaluate on 500 sampled images from this set. As shown in Table 12, The results consistently show GradCAM outperforming Saliency Map, with higher $FFid^+$ (0.1123 vs 0.0672) and lower $FFid^-$ (0.0046 vs 0.0091) scores. It shows that F-Fidelity successfully captures the qualitative differences between explainers, such as GradCAM's more focused attribution maps compared to Saliency Map's noisier explanations.

Table 12: Comparative evaluation of GradCAM and Saliency Map explanations using F-Fidelity on ImageNet dataset with ResNet-18. The $FFid^+$ and $FFid^-$ scores are averaged over different sparsity levels, with bold scores indicating better performance.

| Image | Explanation Visualization | | | Evaluation Metrics | | |
|---|---|---|---|---|---|---|
| | Original | GradCAM | Saliency Map | Method | $FFid^+ \uparrow$ | $FFid^- \downarrow$ |
| Case 1 | | | | GradCAM | **0.3476** | **0.0019** |
| | | | | Saliency Map | 0.2362 | 0.0057 |
| Case 2 | | | | GradCAM | **0.3981** | **0.0076** |
| | | | | Saliency Map | 0.3571 | 0.0105 |
| Case 3 | | | | GradCAM | **0.5438** | **0.0257** |
| | | | | Saliency Map | 0.2162 | 0.0590 |
| Case 4 | | | | GradCAM | **0.6190** | **0.0029** |
| | | | | Saliency Map | 0.1019 | 0.0162 |
| Case 5 | | | | GradCAM | **-0.0010** | **-0.2467** |
| | | | | Saliency Map | -0.1477 | -0.0895 |
| Average Over 500 Images | | | | GradCAM | **0.1123** | **0.0046** |
| | | | | Saliency Map | 0.0672 | 0.0091 |

## D.11 COMPUTATIONAL EFFICIENCY ANALYSIS

Following the comparative setup between GradCAM and Saliency Map on Tiny-Imagenet, we conduct a detailed computational efficiency analysis. Table 13 breaks down the computational time requirements for both ROAR and F-Fidelity. The key advantage of F-Fidelity lies in its one-time fine-tuning cost - while ROAR requires separate explaining and training processes for each explainer (totaling 7140s for two explainers), F-Fidelity needs only a single fine-tuning process (2925s) that can be reused across all explainers.

For the Tiny-Imagenet dataset with ResNet backbone, ROAR's per-explainer cost includes generating explanations for training samples (approximately 650-700s), model retraining (around 2900s), and evaluation (about 120s). In contrast, F-Fidelity requires only a single fine-tuning step (2925s), followed by evaluation for each explainer (approximately 550s). This translates to a total processing time of 4026s for F-Fidelity compared to 7378s for ROAR when evaluating two explainers, with the gap widening further as more explainers are evaluated.

This efficiency gain becomes particularly significant when evaluating multiple explainers or conducting extensive ablation studies, making F-Fidelity more practical for real-world applications while maintaining superior evaluation quality as demonstrated in our previous experiments. The advantage is even more pronounced for computationally intensive explainers, as ROAR requires

Table 13: Computational time analysis for evaluating GradCAM and Saliency Map on Tiny-Imagenet using ResNet as the backbone. While ROAR requires separate explaining and training processes for each explainer, F-Fidelity's fine-tuning cost is one-time.

| Method | Explaining Training Samples (100k) | Model Training/ Fine-tuning (100 epochs) | Evaluation (2000 samples) | Total Time |
|---|---|---|---|---|
| ROAR | 647s + 721s | 2911s + 2861s | 114s+124s | 7378s |
| F-Fidelity | - | 2925s | 541s+560s | 4026s |

generating explanations for the entire training set (100k+ samples in our case), while F-Fidelity only needs explanations for the evaluation set. Moreover, our hyperparameter analysis shows that we can further reduce the evaluation time by decreasing the number of samples - with $N = 10$, the evaluation time per explainer drops from approximately 550s to 180s while maintaining robust performance. This flexibility allows users to balance between evaluation speed and precision based on their specific needs.

# E  DETAILED EXPERIMENTAL RESULTS

## E.1  IMAGE CLASSIFICATION EXPLANATION EVALUATION

**Experiments on CIFAR-100**. We provide the detailed experiment results on the CIFAR-100 dataset. The CIFAR-100 dataset is a collection of 60,000 color images, each sized 32x32 pixels, classified into 100 distinct categories, with 600 images per class. It contains 50,000 training images and 10,000 test images. For the ResNet backbone, in Table 14, we compared different methods on the $Fid+$ and $Fid-$ metrics. From the figures, we found R-Fidelity and F-Fidelity methods have the advantage of distinguishing explanations with larger margin scores.

Similar results can be observed for the ViT backbone. We provide the results in Table 15.

Table 14: Fidelity results on CIFAR-100 dataset with ResNet as the model to be explained.

**Experiments on Tiny-Imagenet**. Similar to the CIFAR-100, we provide the detailed experiment results on Tiny-Imagenet in this section, which is a scaled-down version of the ImageNet dataset, designed for image classification tasks. Tiny-Imagenet contains 200 classes and 500 training images per class, 50 validation images, and 50 test images per class. The image resolution is 64x64. From Table 16 and  17, we observe that compared to R-Fidelity and F-Fidelity, F-Fidelity and R-Fidelity have the best performance.

Table 15: Fidelity results on CIFAR-100 dataset with ViT as the model to be explained.

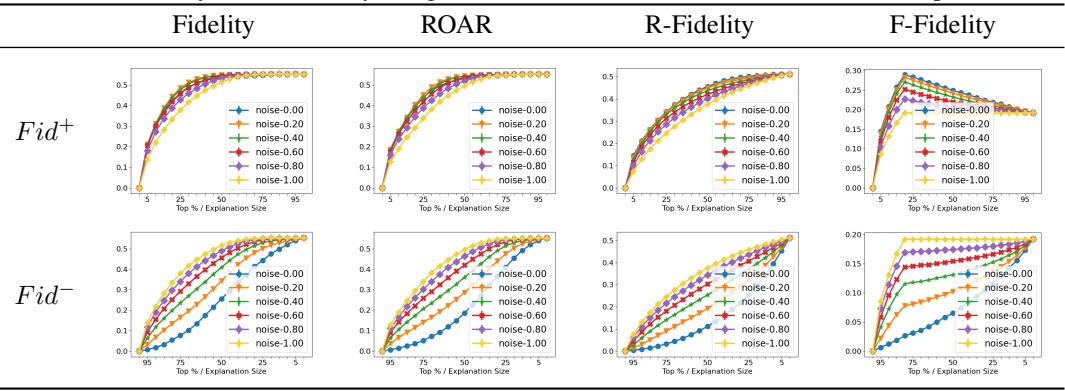

Table 16: Fidelity results on Tiny-Imagenet dataset with ResNet as the model to be explained.

Table 17: Fidelity results on Tiny-Imagenet dataset with ViT as the model to be explained.

## E.2   DETAILED RESULTS ON TIME SERIES DATASETS

We conduct the experiments on the PAM and Boiler datasets with LSTM architecture. As Table 18 and 19 show, our method distinguishes explanations with a larger margin than R-Fidelity. Moreover, Fidelity and ROAR method fail to distinguish different explanations in the Boiler dataset.

Table 18: Fidelity results on PAM dataset with LSTM as the model to be explained.

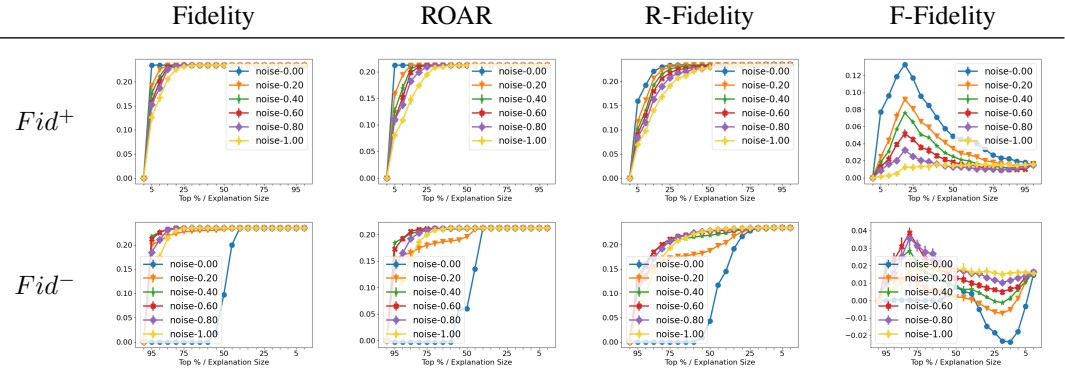

Table 19: Fidelity results on Boiler dataset with LSTM as the model to be explained.

## E.3   DETAILED RESULTS ON NLP

In Table 20, 21, 22 and 23, with LSTM and Transformer, we find the Fidelity has a consistent result which is quite different than other tasks. Compared to R-Fidelity and F-Fidelity, the results are almost the same both in the SST2 and BoolQ datasets, which indicates in these two cases, the OOD problem can be ignored.

Table 20: Fidelity results on SST2 dataset with LSTM as the model to be explained.

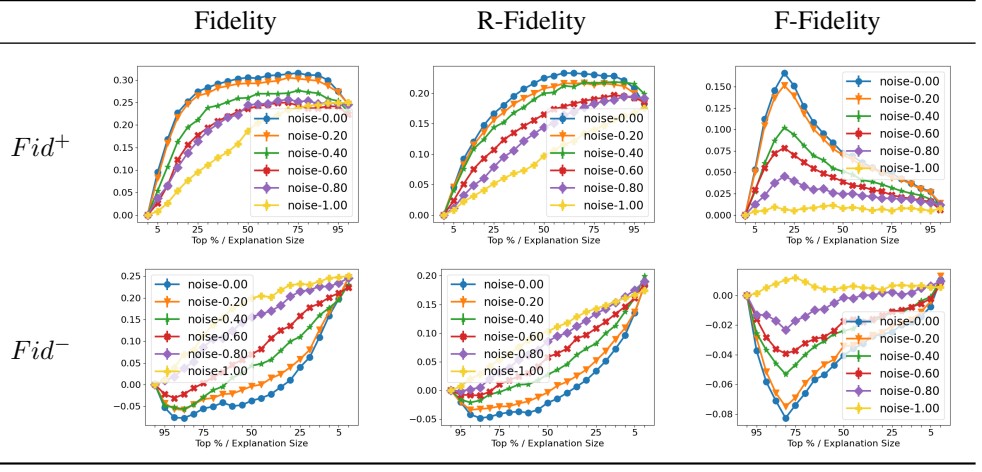

Table 21: Fidelity results on SST2 dataset with Transformer as the model to be explained.

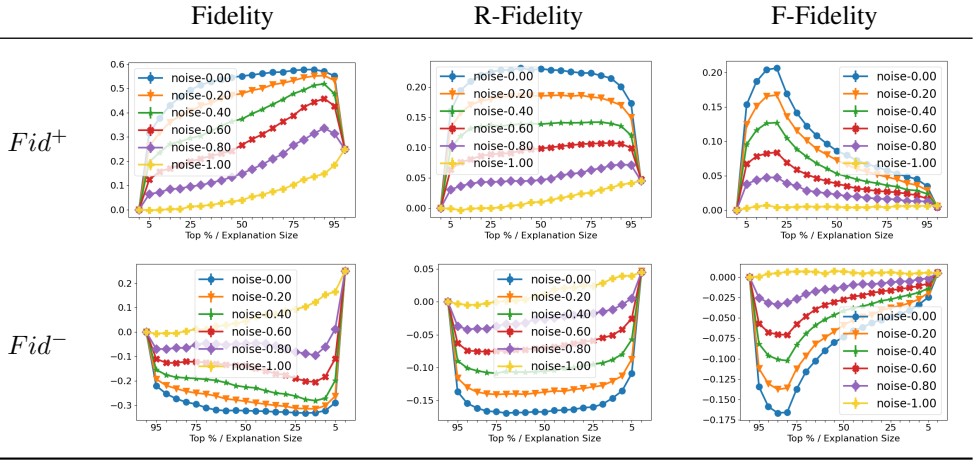

Table 22: Fidelity results on BoolQ dataset with LSTM as the model to be explained.

Table 23: Fidelity results on BoolQ dataset with Transformer as the model to be explained.

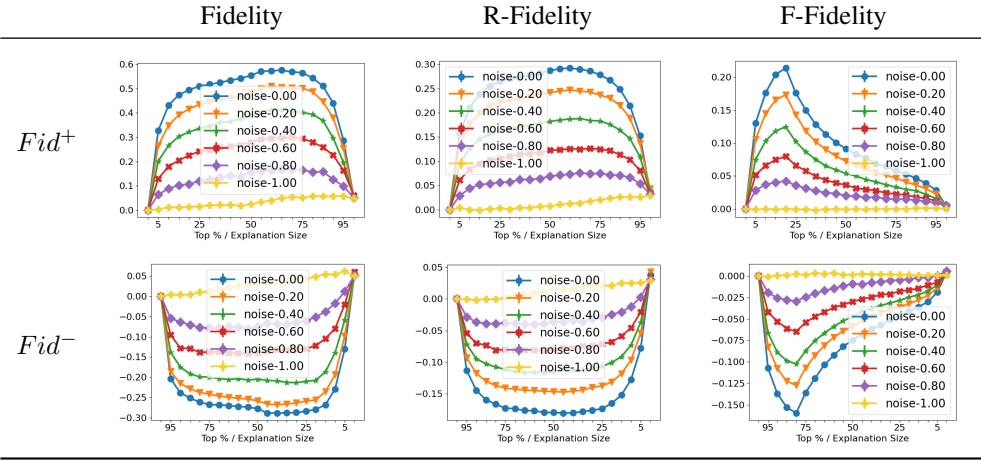

### E.4 DETAILED RESULTS OF HYPERPARAMETER SENSITIVITY STUDY

In this section, we provide the detailed ablation study experiment results on the Colored-Mnist dataset. The experiment settings are described as Section D.9. We provide the detailed $Fid^+$, $Fid^-$, and three Spearman rank Correlation, same as the subsection E.1. As Table 24, 25, and 26 show, our method is robust to the choice of hyperparameters.

Table 24: The detailed ablation study results on sampling number $N$, with $\beta = 0.1$, $\alpha^+ = \alpha^- = 0.5$.

Table 25: The detailed ablation study results on hyperparameter $\beta$, with $N = 50$, $\alpha^+ = \alpha^- = 0.5$.

Table 26: The detailed ablation study results on hyperparameter $\alpha = \alpha^+ = \alpha^-$, with $N = 50$, $\beta = 0.1$.

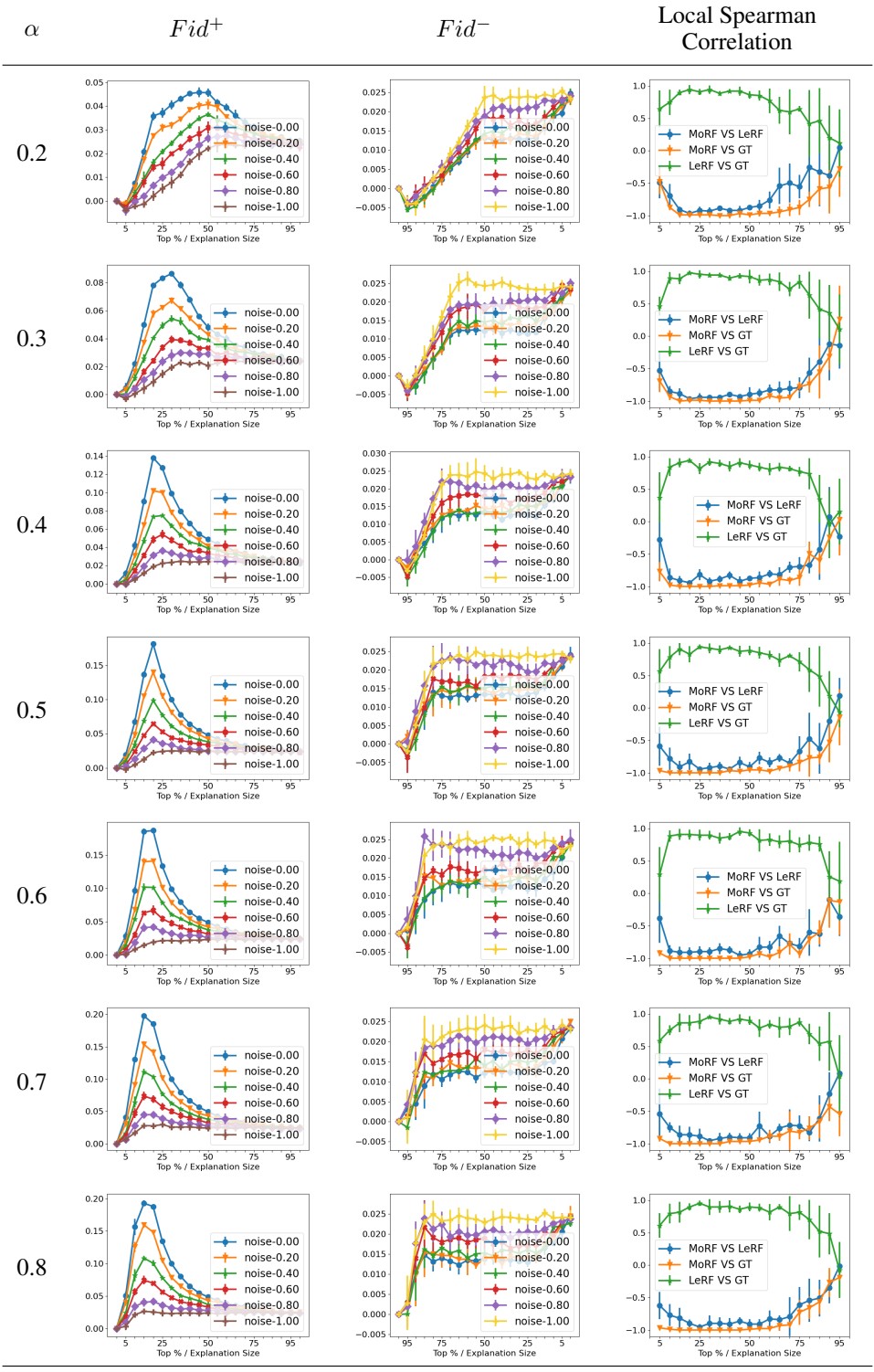

