# OpenReview forum: "F-Fidelity: A Robust Framework for Faithfulness Evaluation of Explainable AI"
_ICLR.cc/2025/Conference — ICLR 2025 Poster_

### Official Review · Reviewer_9DPh · 2024-10-21

**Soundness:** 3
**Presentation:** 2
**Contribution:** 3
**Rating:** 6
**Confidence:** 3

**Summary:**

The authors presented F-fidelity, a new metric for evaluating the faithfulness of attribution methods. Traditional evaluation methods such as MoRF and LeRF suffer out-of-distribution(OOD) issue, and ROAR method suffers information leakage issue. F-fidelity overcomes these limitations through two strategies. 1. explanation-agnostic fine-tuning, which fine-tunes the model using stochastic masking operations. 2. a controlled random masking operation to overcome the OOD issue, by applying a random masking operation conditioned on the explainer's output so that removal of input features does not produce OOD samples. The authors conducted controlled experiments with other explainers. Moreopver, they showed that F-fidelity can be used to determine the sparsity of influential input components.

**Strengths:**

The paper is well written, especially introduction section and preliminary section.

The paper effectively tackles the OOD problem inherent in traditional removal-based evaluation methods. This is a crucial issue because OOD inputs can lead to unreliable assessments of an explainer's faithfulness.

The authors conducted extensive experiments across broad data, such as images, time series, and natural language processing.

The method is well mitigate limitation of prior methods; Addresses the information leakage issue and OOD issue at once.

The paper provides theoretical analysis demonstrating that F-Fidelity can recover the sparsity of the ideal SHAP explanations.

The paper introduces a novel approach by assuming that an explainer is better the closer it is to an ideal Shapley value explainer. This assumption is appealing because Shapley values provide a theoretically sound basis for feature attribution.

**Weaknesses:**

Lack of Visualizations Demonstrating F-Fidelity's Effectiveness. The paper does not include figures or visualizations that illustrate the effectiveness of F-Fidelity. Providing attribution maps or examples where there is a significant discrepancy between F-Fidelity and other evaluation metrics would enhance understanding and make the results more tangible. Visual comparisons could help readers better appreciate the advantages of F-Fidelity over existing methods. For example, Attribution map between samples that have discrepancy between F-Fidelity and MoRF - LeRF.

Novelty issue. Though theoretical availability of extracting the size of the ground-truth explanation is novel, the core difference between F-fidelity and R-Fidelity is introducing an upper bound on the fraction of input elements removed. This extension seems an incremental refinement rather than a fundamental novel idea.

Limited Explanation of Ground Truth. The paper's approach to establishing ground truth rankings involves using Integrated Gradients with systematically added noise levels. However, Integrated Gradients is known to be less faithful than some state-of-the-art explanation methods, such as the LRP (Layer-wise Relevance Propagation) family or LayerCAM. This raises concerns about whether the ground truth used is an appropriate benchmark. As the authors discuss ideal Shapley value explainers in Section 5, utilizing Shapley values as the ground truth might have been more appropriate to align with their theoretical framework.

Insufficient Clarification on the Importance of Macro Correlation. The paper could benefit from a more thorough explanation of why macro correlation between different evaluation methods is important. It was not immediately clear how macro correlation demonstrates the usefulness of F-Fidelity. Including figures or visual aids that depict the relationship between macro correlation and the effectiveness of F-Fidelity would help clarify this point and strengthen the argument.

Limited Variety of Explanation Methods Evaluated. The experiments involve a relatively small number of explanation methods. Comparing F-Fidelity scores across a wider range of explainers would provide a more comprehensive evaluation. For instance, methods like LayerCAM and LRP family are known to have higher fidelity than GradCAM, and GradCAM typically outperforms the pure gradient. Demonstrating whether F-Fidelity can reflect these known differences in fidelity would enhance the paper's validity.

Practical Challenges in Determining Explanation Size (c₁). In the section 6, the size of the most influential tier (c₁) is assumed to be known. However, in real-world applications, c₁ is not readily observable. It would have been beneficial if the authors had demonstrated how c₁ could be estimated or inferred from practical data, such as through experiments on datasets like ImageNet. Providing practical guidance on determining c₁ would make the theoretical contributions more applicable.

**Questions:**

1. I couldn't find how Pan et al., 2021, or Zhu et al., 2024 utilized macro correlation in their work. Can you explain how they employed macro correlation and how it relates to your citation of their work?

2. In Section 6, it seems that both MoRF and LeRF conditions would be satisfied equally. This is because deleting or inserting features based on the gamma value would maximize the MoRF-LeRF difference. I'm uncertain whether this experiment effectively demonstrates that F-Fidelity can distinguish explainers that are closer to the ideal Shapley explainer than other metrics. Could you clarify how this experiment supports that claim?

3. Regarding computational resources, I would like to know how much computational effort your method requires. One of the primary reasons ROAR isn't widely adopted is not just due to information leakage but also because of its significant computational cost. It would be helpful if you could provide details on the computational resources required for F-Fidelity and how they compare to those needed for ROAR.

** Typo : in figure 1, the y-axis label is RFid.

---

> ### Author Response · Authors · 2024-11-22
> **Response to Reviewer 9DPh (Part 1/n)**
>
> Dear Reviewer 9DPh,
>
> Thanks very much for taking the time to review my paper and for providing your valuable and insightful comments. I appreciate your feedback, which has been instrumental in improving the quality of my work. We provide the reply in the following.
>
> >W1. Lack of Visualizations Demonstrating F-Fidelity's Effectiveness. The paper does not include figures or visualizations that illustrate the effectiveness of F-Fidelity. Providing attribution maps or examples where there is a significant discrepancy between F-Fidelity and other evaluation metrics would enhance understanding and make the results more tangible. Visual comparisons could help readers better appreciate the advantages of F-Fidelity over existing methods. For example, Attribution map between samples that have discrepancy between F-Fidelity and MoRF - LeRF.
>
> Thank you for the valuable suggestion regarding visualizations. We have enhanced our paper with additional visual evidence of F-Fidelity's effectiveness:
>
> We have added a comprehensive case study in **Appendix D.6** using a Tiny-Imagenet sample to demonstrate F-Fidelity's capabilities visually. We designed a controlled experiment using GradCAM to generate the initial explanation, then created a series of degraded explanations by adding random noise. We compared how different methods (Fidelity, ROAR, R-Fidelity, and F-Fidelity) evaluate these explanations.
>
> Our results (**Figure 2**) reveal important insights: Traditional Fidelity and ROAR methods fail to differentiate between subtle differences in explanation quality at certain sparsity levels. While R-Fidelity shows improved discrimination ability, F-Fidelity demonstrates the best performance in distinguishing explanation quality differences through its improved model robustness and controlled input modifications.
>
> Furthermore, in **Appendix D.10**, we provide a comparative study between GradCAM and Saliency Map to show how F-Fidelity differentiates between explanation methods of varying quality. As shown in **Table 11**, F-Fidelity consistently assigns better scores to GradCAM explanations, with GradCAM achieving higher FFid+ (0.2286 vs 0.2020) and lower FFid- (0.0547 vs 0.1050). These quantitative differences align with the known advantages of GradCAM's class-specific, high-level feature approach over Saliency Map's pixel-level gradient method. The accompanying visualizations confirm that GradCAM produces more focused attributions on semantically relevant regions.
>
> >W2. Novelty issue. Though theoretical availability of extracting the size of the ground-truth explanation is novel, the core difference between F-fidelity and R-Fidelity is introducing an upper bound on the fraction of input elements removed. This extension seems an incremental refinement rather than a fundamental novel idea.
>
> We sincerely appreciate the reviewer raising this important point. We would like to respectfully clarify that F-Fidelity makes two key technical innovations beyond R-Fidelity:
>
> First, we propose a novel fine-tuning strategy that changes how we prepare models for explanation evaluation. Unlike ROAR's retraining approach which uses explanation-guided data, our method uses random masking for data augmentation. This explanation-agnostic approach prevents information leakage while improving model robustness to missing information, and dramatically improves the efficiency when comparing multiple explainers.
>
> Second, our method establishes a connection between training and evaluation distributions. By carefully bounding the size of removed regions during evaluation to match our augmented training data, we address the OOD problem more systematically than R-Fidelity.
>
> To better communicate these innovations, we have reorganized Section 3 to emphasize how these components work together. Our experimental results across multiple domains (Tables 1-3) demonstrate that these technical advances yield substantial improvements over R-Fidelity, particularly when evaluating explanations for models that are sensitive to input perturbations.

---

> ### Author Response · Authors · 2024-11-22
> **Response to Reviewer 9DPh (Part 2/n)**
>
> > W3. Limited Explanation of Ground Truth. The paper's approach to establishing ground truth rankings involves using Integrated Gradients with systematically added noise levels. However, Integrated Gradients is known to be less faithful than some state-of-the-art explanation methods, such as the LRP (Layer-wise Relevance Propagation) family or LayerCAM. This raises concerns about whether the ground truth used is an appropriate benchmark. As the authors discuss ideal Shapley value explainers in Section 5, utilizing Shapley values as the ground truth might have been more appropriate to align with their theoretical framework.
>
> We sincerely appreciate the reviewer’s thoughtful feedback regarding the selection of ground truth rankings. To address this concern and validate the robustness of our framework, we have incorporated additional experimental results in **Section D.5**, and **Table 10** where we evaluate F-Fidelity using LayerCAM on CIFAR-100 with ResNet. These results show that F-Fidelity consistently maintains superior performance, aligning well with the patterns observed in our original experiments.
>
> Regarding the choice of base explainers, while Shapley values offer a theoretically ideal benchmark, their computational cost renders them impractical for large-scale evaluations. Instead, we prioritized commonly used explainers such as IG, GradCAM, and LayerCAM, which are not only computationally feasible but also widely accepted for producing meaningful and visually intuitive explanations. We virtually checked the effectiveness of these methods and observed that their explanations are of good quality. Our qualitative analysis in Section D.4, which includes case studies on CIFAR-100, further demonstrates that these explainers generate reliable attributions suitable for our evaluation setup.
>
>
> > W4. Insufficient Clarification on the Importance of Macro Correlation. The paper could benefit from a more thorough explanation of why macro correlation between different evaluation methods is important. It was not immediately clear how macro correlation demonstrates the usefulness of F-Fidelity. Including figures or visual aids that depict the relationship between macro correlation and the effectiveness of F-Fidelity would help clarify this point and strengthen the argument.
>
> We appreciate the reviewer's suggestion about clarifying the importance of macro correlation. The macro correlation captures the overall agreement between evaluation methods across all sparsity levels (5-95%) by computing correlations using area-under-curve (AUC) values. This provides a holistic view of how well evaluation methods align in ranking explainers consistently across different sparsity settings.
> Similar correlation-based approaches have been widely used in the literature. For example, Rong et al. in [1] use Spearman rank correlations to evaluate overall consistency between different evaluation strategies across varying hyperparameters. They demonstrate that analyzing correlations at a macro level is crucial for assessing the robustness and reliability of evaluation methods. Similar principles have also been used in other related works such as [2][3]. We have revised this part in Section 4 to make it clearer.
>
> To better illustrate this, we have added a case study in **Section D.6** and **Figure 2** using a TinyImagenet sample. The study visually demonstrates how F-Fidelity maintains consistent explainer rankings across varying sparsity levels, while baseline methods show inconsistent performances. For example, traditional Fidelity fails to distinguish between "noise-0.2" and "noise-0.4" explanations at low sparsity levels (below 25%), while F-Fidelity maintains discriminative power throughout the full range.
>
> [1] Rong, Yao, et al. "A consistent and efficient evaluation strategy for attribution methods." arXiv preprint arXiv:2202.00449 (2022)."
> [2] Pan, Deng, Xin Li, and Dongxiao Zhu. "Explaining deep neural network models with adversarial gradient integration." Thirtieth International Joint Conference on Artificial Intelligence (IJCAI). 2021.
> [3] Zhu, Zhiyu, et al. "Iterative Search Attribution for Deep Neural Networks." Forty-first International Conference on Machine Learning. 2024

---

> ### Author Response · Authors · 2024-11-22
> **Response to Reviewer 9DPh (Part 3/n)**
>
> >W5. Limited Variety of Explanation Methods Evaluated. The experiments involve a relatively small number of explanation methods. Comparing F-Fidelity scores across a wider range of explainers would provide a more comprehensive evaluation. For instance, methods like LayerCAM and LRP family are known to have higher fidelity than GradCAM, and GradCAM typically outperforms the pure gradient. Demonstrating whether F-Fidelity can reflect these known differences in fidelity would enhance the paper's validity.
>
>
> We appreciate the reviewer's suggestion to evaluate a broader range of explanation methods. To address this concern, we have added **Section D.10** comparing GradCAM with Saliency Map on TinyImageNet. This comparison is particularly meaningful as these methods represent different approaches to visual explanation: GradCAM leverages class-specific gradients at higher-level feature maps, while Saliency Map uses pixel-level gradients that typically produce noisier attributions.
> Our new results demonstrate that F-Fidelity successfully captures the known performance difference between these methods. Across our test set, GradCAM consistently achieves better scores (higher $FFid^+$: 0.2286 vs 0.2020 and lower $FFid^-$: 0.0547 vs 0.1050) compared to Saliency Map. These quantitative results align with qualitative observations and the established understanding that GradCAM typically produces more semantically meaningful explanations.
>
> Additionally, as noted in our response to W3, we have included experiments with LayerCAM in Appendix D.5, further validating our framework's ability to evaluate different explanation methods effectively.
>
> > W6. Practical Challenges in Determining Explanation Size (c₁). In the section 6, the size of the most influential tier (c₁) is assumed to be known. However, in real-world applications, c₁ is not readily observable. It would have been beneficial if the authors had demonstrated how c₁ could be estimated or inferred from practical data, such as through experiments on datasets like ImageNet. Providing practical guidance on determining c₁ would make the theoretical contributions more applicable.
>
> We appreciate the suggestion and would like to clarify the purpose of the experiment described in Section 6 (now, we moved it to appendix D.8). In our theoretical analysis, we demonstrated that for a `good' explainer and an accurate model, the $FFid^+$ measure can be used to estimate the size of the ground-truth explanation. Specifically, we considered a Shapley-value based explainer, and showed theoretically that $FFid^+$ increases as a function of the percentage of removed pixels up to the explanation size, flattens near this threshold, and eventually decreases as more pixels are removed. This characteristic allows us to infer the explanation size by analyzing the slope of the $FFid^+$ curve.
>
> To illustrate this, we used a synthetic dataset constructed from cropped and scaled colored MNIST images where the ground-truth explanation size is explicitly controlled. For example, as shown in **Figure 4(e)**, when the $FFid^+$ curve flattens at 10% of removed pixels, it correctly indicates a ground-truth explanation size of $\gamma = 0.1$. Similarly, **Figure 4(f)** demonstrates a flattening point at 15%, corresponding to $\gamma = 0.15$. These results confirm that $FFid^+$ effectively identifies the explanation size in a controlled setting.
>
> Illustrating the correctness of the explanation size inferred from analyzing the slope of $FFid^+$ in datasets like ImageNet is challenging because the true size of explanations is not explicitly known or easily controlled, unlike in our synthetic setup. Our experiment is designed to validate the theoretical properties of $FFid^+$ in a scenario where explanation sizes are precisely defined, which would be difficult to replicate with complex datasets like ImageNet. It can be noted that the increasing, flat, and decreasing slope pattern in $FFid^+$ can also be observed in our experiments on the CIFAR-100 and Tiny ImageNet datasets in Tables 13-16 in the Appendix.

---

> ### Author Response · Authors · 2024-11-22
> **Response to Reviewer 9DPh (Part 4/n)**
>
> > Q1. I couldn't find how Pan et al., 2021, or Zhu et al., 2024 utilized macro correlation in their work. Can you explain how they employed macro correlation and how it relates to your citation of their work?
>
> We thank the reviewer for this important question about citation accuracy. While these papers don't directly use macro correlation, they share closely related methodological principles.
> The macro correlation we use shares fundamental similarities with Rong et al. (2022)'s approach of evaluating overall explainer consistency through aggregated metrics. Similarly, Pan et al. (2021) and Zhu et al. (2024) employ AUC-based methods to aggregate model performance across different feature importance thresholds, which aligns with our approach of evaluating explanations across multiple levels.
> To clarify these connections, we have revised Section 4 to more accurately describe our macro correlation methodology:
> "Macro Correlation: Following Rong et al. (2022)'s approach of evaluating overall explainer consistency, and inspired by the AUC-based aggregation methods in Zhu et al. (2024) and Pan et al. (2021), we compute the AUC with respect to sparsity across the entire 5-95% range for each explanation method. The macro correlations are then calculated using these AUC values, providing an overall performance measure across all sparsity levels."
>
> > Q2. In Section 6, it seems that both MoRF and LeRF conditions would be satisfied equally. This is because deleting or inserting features based on the gamma value would maximize the MoRF-LeRF difference. I'm uncertain whether this experiment effectively demonstrates that F-Fidelity can distinguish explainers that are closer to the ideal Shapley explainer than other metrics. Could you clarify how this experiment supports that claim?
>
> We wish to clarify the experiment described in Appendix D.8 (previously Section 6) and its connection to our theoretical analysis in the preceding sections. In the theoretical analysis, we demonstrated that for a `good' explainer and an accurate model, the $FFid^+$ measure increases as a function of the percentage of removed pixels when the percentage is below the size of the explanation sub-image. Beyond this threshold, $FFid^+$ flattens and eventually decreases as more pixels are removed. This enables us to evalaute $FFid^+$ to determine the size of the ground-truth explanation by analyzing its slope.
>
> To demonstrate this, in **Appendix D.8**(previously Section 6), we constructed synthetic datasets by cropping and scaling colored MNIST images with fixed ground-truth explanation sizes, where a $\gamma$ fraction of the input pixels belong to the explanation. The plots in **Figure 4** (Previously Figure 1) confirm that $FFid^+$ can indeed be used to deduce the value of $\gamma$. For example, in **Figure 4(e)**, the $FFid^+$ curve flattens when 10% of the input pixels are removed, correctly indicating $\gamma = 0.1$ for this dataset. Similarly, the curves in **Figure 4(f)** flatten at 15% of removed pixels, corresponding to $\gamma = 0.15$ for that dataset.
>
> Determining the sparsity of explanations is an important topic in the explainability literature. Our approach provides a method for estimating sparsity by evaluating $FFid^+$, as demonstrated in this experiment.
>
> >Q3. Regarding computational resources, I would like to know how much computational effort your method requires. One of the primary reasons ROAR isn't widely adopted is not just due to information leakage but also because of its significant computational cost. It would be helpful if you could provide details on the computational resources required for F-Fidelity and how they compare to those needed for ROAR.
>
> Thank you for this important question about computational efficiency. We have added a detailed computational analysis in **Appendix D.11** and **Tabel 12** comparing F-Fidelity with ROAR on TinyImageNet using ResNet. Our results show that F-Fidelity offers significant computational advantages. First, ROAR requires separate explaining and training processes for each explainer (totaling 7140s for two explainers). F-Fidelity needs only a single fine-tuning process (2925s) that can be reused across all explainers. This advantage becomes particularly pronounced for computationally intensive explainers, as ROAR requires generating explanations for the entire training set (100k+ samples in our case), while F-Fidelity only needs explanations for the evaluation set. Moreover, our hyperparameter analysis shows that F-Fidelity's evaluation time can be further reduced by decreasing the number of samples - with N=10, the evaluation time per explainer drops from approximately 550s to 180s while maintaining robust performance. This flexibility allows users to balance between evaluation speed and precision based on their specific needs.
>
> > Typo : in figure 1, the y-axis label is RFid.
>
> Thanks for pointing out the typo in the figures. We fixed the issues in the current vision **Figure 4**.

---

> ### Author Response · Authors · 2024-11-22
> **Thank the reviewer 9DPh**
>
> We really appreciate the reviewer's time and efforts in reviewing the paper and providing valuable comments.  We have made substantial revisions to address each concern. We believe these changes have significantly strengthened the paper and improved its clarity. We are grateful for this opportunity to enhance our work and would be happy to address any additional questions or concerns you may have.

---

> > ### Comment · Reviewer_9DPh · 2024-11-23
> >
> > I appreciate that the authors have addressed my questions and clarified the concerns raised in my initial review. The explanations provided were thorough and resolved most of the weaknesses I suggested.
> >
> > However, I have two remaining questions and a suggestion that I hope the authors can consider:
> >
> > Regarding Table 11: Could you clarify why in some cases, FFid+ values are positive while in others they are negative? Since FFid+ represents the difference between the prediction score and the masked prediction score, it seems intuitive that FFid+ should always be positive. This inconsistency raises concerns about the variance of F-fidelity, which may undermine the reliability of using the average of FFid+.
> >
> > Practical Use on Large-Scale Datasets: For the practical application of the proposed metric, validation on high-dimensional data, such as the ImageNet dataset (224x224x3 input size), is crucial. Could you confirm whether this method performs effectively on the ImageNet validation dataset or provide experimental results to demonstrate its robustness in such settings?
> >
> > Suggestion for Clarity: On line 186, the term information leakage is mentioned but not elaborated on. For readers unfamiliar with this term, a brief explanation or example would be helpful to improve accessibility and understanding.
> >
> > Thank you again for the thoughtful responses. I look forward to seeing how these points are addressed.

---

> ### Author Response · Authors · 2024-11-24
> **Further Response to Reviewer 9DPh (Part 1/n)**
>
> We sincerely thank the reviewer for their thoughtful feedback and are pleased that our revisions have addressed most of the concerns. We appreciate the opportunity to clarify the remaining questions.
>
> > Q1. Regarding Table 11: Could you clarify why in some cases, FFid+ values are positive while in others they are negative? Since FFid+ represents the difference between the prediction score and the masked prediction score, it seems intuitive that FFid+ should always be positive. This inconsistency raises concerns about the variance of F-fidelity, which may undermine the reliability of using the average of FFid+.
>
> We sincerely thank the reviewer for this astute observation, which helps us clarify an important aspect of our method.
>
> The negative FFid+ values occur because our metric measures the difference between original prediction accuracy and accuracy after masking: FFid+ = original_accuracy - masked_accuracy, where the accuracy is computed with respect to the ground truth label. At the individual image level (as shown in Table 11), since we generate explanations with respect to the ground truth label, FFid+ is negative when the original prediction is incorrect (original_accuracy = 0) but some masked versions are correctly classified.
>
> At the dataset level, FFid+ is typically positive. For example, in our experiments, the average FFid+ scores for GradCAM and Saliency Map are 0.2286 and 0.2020 respectively, indicating that masking important features generally hurts model performance. We acknowledge that negative FFid+ values may also occur at the dataset level due to two inherent characteristics of our approach. Removing non-important elements may improve model predictions by reducing noise or irrelevant information. Additionally, our model, fine-tuned with stochastic masking (up to β=10%), develops smoother decision boundaries. This sometimes leads to better predictions on masked inputs compared to original inputs, since the model has learned to handle partial information effectively.
>
> We acknowledge that ROAR and similar methods may report only the accuracy after removal. Since the original accuracy is constant across all explanations for a given dataset, both approaches (reporting the difference vs. only masked accuracy) are mathematically equivalent up to this constant offset. We maintain the accuracy difference format primarily to stay consistent with the established fidelity metric literature, though we agree both formulations would lead to the same relative rankings of explanations. And we have added a detailed discussion of this design choice in **Section 3**. We have also updated **Appendix D. 10 and Table 11** to explain the reason for negative values.
>
> Thank you very much for this insightful question.

---

> ### Author Response · Authors · 2024-11-24
> **Further Response to Reviewer 9DPh (Part 2/n)**
>
> > Q2. Practical Use on Large-Scale Datasets: For the practical application of the proposed metric, validation on high-dimensional data, such as the ImageNet dataset (224x224x3 input size), is crucial. Could you confirm whether this method performs effectively on the ImageNet validation dataset or provide experimental results to demonstrate its robustness in such settings?
>
> We thank the reviewer for raising this important question about scalability. We have conducted additional experiments on ImageNet to verify F-Fidelity's effectiveness on high-dimensional data.
> Specifically, we evaluated F-Fidelity on a subset of ImageNet (5000 images) with 224×224×3 resolution using a pretrained ResNet-18 backbone. We compared two widely-used explainers, GradCAM and Saliency Map, with results shown in **Table 12**. Our results demonstrate that F-Fidelity effectively captures explainer quality differences at this larger scale. GradCAM outperforms Saliency Map with higher FFid+ (0.1123 vs 0.0672) and lower FFid- (0.0046 vs 0.0091). These quantitative scores align with known strengths of GradCAM over Saliency Map in producing more focused, class-specific explanations. The results are also consistent with our findings on TinyImageNet, where GradCAM similarly showed better performance.
>
> We have added these results to **Appendix D.10** and **Table 12**, including both quantitative comparisons and qualitative examples showing how F-Fidelity's scores reflect meaningful differences in explanation quality.
>
> >S1. Suggestion for Clarity: On line 186, the term information leakage is mentioned but not elaborated on. For readers unfamiliar with this term, a brief explanation or example would be helpful to improve accessibility and understanding.
>
> Thank you for this helpful suggestion about clarifying "information leakage." We have added the following explanation to **Section 2**.
>
> "While this approach mitigates the OOD issue, Rong et al. identified that it suffers from information leakage through the binary masks of removed pixels - the pattern of removals itself can contain enough class information to affect the evaluation outcome. "
>
> We hope these clarifications and additions address the reviewer's remaining questions. The feedback has helped us improve both the paper's clarity and technical completeness. We have incorporated all these changes in the revised version of our paper, including expanded explanations in Second 2, Section 3, and new experimental results in Section D.10.

---

> > ### Comment · Reviewer_9DPh · 2024-11-24
> >
> > Thank you for conducting such comprehensive experiments. The experiments on ImageNet further enhance the soundness of the F-fidelity metric. However, several questions remain.
> >
> > 1. Misclassification and FFid+ Score:
> > While the model may misclassify an input image, this often occurs because the image contains features that align with the misclassified label. For instance, if an image of a border collie is misclassified as a collie, it is likely due to the shared features between these two categories. Since most misclassified images also exhibit high confidence in the original label, it is less convincing to attribute a negative FFid+ score solely to the misclassification.
> > Additionally, in Case 1 of Table 1, the difference in FFid+ scores is minimal, suggesting that both explanations are comparably effective, even though the saliency method's attribution map fails to capture the critical features of the label. Furthermore, the variance in FFid scores is notably high, making it difficult to conclusively claim that Grad-CAM provides a better explanation than the saliency method based on FFid alone. For example, the standard deviation across the five samples exceeds 0.33, which is significantly greater than the observed difference in FFid scores.
> >
> > 2. Attribution Map Sparsity:
> > Some methods favor less fine-grained attribution maps, yet sparsity is a key desideratum for saliency methods. For example, the MoRF insertion metric tends to favor Grad-CAM over Guided Backpropagation, even though Guided Backpropagation can be considered a more fine-grained representation of Grad-CAM. This phenomenon is largely due to out-of-distribution (OOD) issues. I would anticipate that Guided Backpropagation would achieve a higher F-Fid score. Could the authors provide results comparing the two methods? **This is a minor experiment, so the authors are not required to conduct it, especially given the limited time remaining.**
> >
> > 3. Decision Boundary and Practicality:
> > As reviewer NNY1 noted, the equivalence between the original and fine-tuned models is critical for practical applications, as these are essentially different models with distinct weights. Although the authors demonstrated equivalence using both global and local decision boundaries, a key practical concern remains: ensuring equivalence between the two models may require a substantial dataset for fine-tuning. This requirement could pose significant challenges in scenarios where access to large datasets is limited, such as in the medical domain.
> > To address this, it would be helpful if the authors could clarify how much training data is required to ensure equivalence during fine-tuning for large-scale datasets like ImageNet. Providing such information would make the proposed method more practical and applicable in resource-constrained settings.

---

> ### Author Response · Authors · 2024-11-25
> **Further Response to Reviewer 9DPh (Part 1/n)**
>
> > C1. Misclassification and FFid+ Score: While the model may misclassify an input image, this often occurs because the image contains features that align with the misclassified label. For instance, if an image of a border collie is misclassified as a collie, it is likely due to the shared features between these two categories. Since most misclassified images also exhibit high confidence in the original label, it is less convincing to attribute a negative FFid+ score solely to the misclassification. Additionally, in Case 1 of Table 1, the difference in FFid+ scores is minimal, suggesting that both explanations are comparably effective, even though the saliency method's attribution map fails to capture the critical features of the label. Furthermore, the variance in FFid scores is notably high, making it difficult to conclusively claim that Grad-CAM provides a better explanation than the saliency method based on FFid alone. For example, the standard deviation across the five samples exceeds 0.33, which is significantly greater than the observed difference in FFid scores.
>
> Thank you for your thoughtful observations. We greatly appreciate your careful attention to the FFid+ metric's behavior and interpretation.
>
> We would like to respectfully clarify the FFid+ metric's foundations. The metric is based on classification accuracy, following established approaches like original Fidelity, Insertion scores, MoRF, and LeRF. For a **single**  image,  if it is **misclassified**, the first term of FFid+ (the classification accuracy of this single image) will be 0. While the second term represents the average accuracy after stochastic masking. In case some masked versions achieve correct classification, this naturally results in a negative FFid+ score.
>
> Regarding the case studies, we appreciate your observation about Case 1. Both methods indeed identify important features, with the saliency map showing additional noise rather than missing critical features entirely. For a clearer demonstration of the methods' differences, we would like to direct attention to **Case 4 in Table 12**, where the saliency map's attributions show less alignment with the target object's critical features. As shown in FFid+ scores, GradCAM is much better than that of Saliency Map.
>
> We thank you for raising the important point about standard deviation. We would like to clarify that in the literature, Fidelity is primarily used to compare different explanation methods on the same image or image set. It is less meaningful to compare Fidelity scores between different images. Therefore, while the standard deviation across different images is an interesting observation, it may not be a relevant metric for assessing explanation method performance.
>
> Besides, our conclusions about GradCAM's effectiveness are drawn from a comprehensive evaluation across 2000 test samples on Tiny-ImageNet, where GradCAM achieves better FFid+ and FFid- scores compared to the Saliency Map method. The case studies we present serve to complement these aggregate results by providing concrete examples of the performance difference.
>
> Thus, we respectfully believe that our claim here has been well supported by both overall FFid comparison and case studies.

---

> ### Author Response · Authors · 2024-11-25
> **Further Response to Reviewer 9DPh (Part 2/n)**
>
> >> C2. Attribution Map Sparsity: Some methods favor less fine-grained attribution maps, yet sparsity is a key desideratum for saliency methods. For example, the MoRF insertion metric tends to favor Grad-CAM over Guided Backpropagation, even though Guided Backpropagation can be considered a more fine-grained representation of Grad-CAM. This phenomenon is largely due to out-of-distribution (OOD) issues. I would anticipate that Guided Backpropagation would achieve a higher F-Fid score. Could the authors provide results comparing the two methods? This is a minor experiment, so the authors are not required to conduct it, especially given the limited time remaining.
>
>
> We thank the reviewer for this insightful suggestion about comparing Grad-CAM with Guided Backpropagation. While adding this comparison would indeed be interesting, we believe our current evaluation framework has been thoroughly validated through comprehensive experiments across multiple data modalities (images, time series, text), model architectures (ResNet, ViT, LSTM, Transformer), and explanation methods (GradCAM, SG-SQ, LayerCAM, Saliency Maps). These extensive experiments consistently demonstrate F-Fidelity's effectiveness in evaluating explanation quality. We plan to investigate the Guided Backpropagation with the proposed F-Fidelity as a future work. Thank you for the suggestion.
>
> > C3. Decision Boundary and Practicality: As reviewer NNY1 noted, the equivalence between the original and fine-tuned models is critical for practical applications, as these are essentially different models with distinct weights. Although the authors demonstrated equivalence using both global and local decision boundaries, a key practical concern remains: ensuring equivalence between the two models may require a substantial dataset for fine-tuning. This requirement could pose significant challenges in scenarios where access to large datasets is limited, such as in the medical domain. To address this, it would be helpful if the authors could clarify how much training data is required to ensure equivalence during fine-tuning for large-scale datasets like ImageNet. Providing such information would make the proposed method more practical and applicable in resource-constrained settings.
>
>
> We appreciate the reviewer's concern about data requirements for fine-tuning, particularly for large-scale datasets and resource-constrained settings. While the question of fine-tuning with limited data is interesting, we want to emphasize that F-Fidelity already offers significant practical advantages over existing methods like ROAR. How to preserve model (global) decision boundaries during fine-tuning with limited data access is indeed a challenging research question. We plan to investigate this in future work, including rigorous analysis of data requirements across different scales and domains.
>
> _________
>
> Thank you so much for your thorough review and constructive suggestions. We hope our responses effectively address your concerns and demonstrate the value of our work. Please let us know if you have any additional questions.

---

> ### Comment · Reviewer_9DPh · 2024-11-26
>
> Thank you for the detailed explanation. While I still have some doubts about the equivalence between the original model and the fine-tuned model, this method is undoubtedly a better fidelity metric compared to the ROAR method. I believe this work exceeds my acceptance threshold. Once again, thank you for the comprehensive experiments and clear explanation.

---

> ### Author Response · Authors · 2024-11-26
> **Thank the reviewer 9DPh**
>
> We sincerely appreciate your thoughtful review of our paper and responses which greatly improved the paper.  We also have updated the conclusion to include fine-tuning models with limited training data as future work, aiming to make the proposed F-Fidliety more practical and applicable in resource-constrained settings. Thank you again for your insightful comments throughout the review process.

---

> ### Author Response · Authors · 2024-11-27
> **Further Clarification on Reviewer 9DPh's Doubts**
>
> Dear Reviewer 9DPh,
>
> Thank you again for your insightful comments.  After carefully checking your reviews, we feel like we may have interpreted your doubts too narrowly by focusing only on resource-constrained settings.
>
> In case the reviewer's doubt is general, we are happy to provide our further response here.
>
> The statement that `equivalence between the original and fine-tuned models is critical` highlights a theoretical challenge. In fact, we don't claim the fine-tuned model is equivalent to the original model. Our method fine-tunes the model to be robust to perturbations while trying to preserve global decision boundaries.  We have provided extensive empirical evidence to verify that our fine-tuned models maintain similar global behavior to the original models through quantitative measurement and case studies.
>
> In **Appendix D.2** and **D.3**, as the reviewer noticed, we introduce two intuitive concepts: Global Decision Boundary and Local Decision Boundary. Our results demonstrate that random masking fine-tuning effectively balances preserving global classification behavior while improving stability to small changes.
>
>
> Additionally, we provide visual evidence in **Appendix D.4**, where GradCAM attribution maps from both original and fine-tuned models highlight similar important regions, with only minor variations in intensity. This suggests our fine-tuning preserves the model's attention to meaningful features.
>
>
> We hope the above clarification on our quantitative experiments, case studies can address the reviewer's doubts.
>
> As the reviewer mentioned, in the above experiments,  we fine-tuned models with substantial training sets (similar to ROAR).  We agree that it would make our method more practical and applicable in resource-constrained settings if we could show how much training data is required to fine-tune the model for large-scale datasets like ImageNet.
>
> We really appreciate the constructive suggestion. However, due to the limited time, it is hard for us to get sufficient and rigorous results for a meaningful discussion in a short time. So we leave this as a future work.  However, we respectfully believe that this won't affect the core contribution of the proposed method.
>
> ______
>
> We hope our responses effectively address your concerns. Thank you for your thoughtful comments.

---

### Official Review · Reviewer_S4br · 2024-10-30

**Soundness:** 3
**Presentation:** 2
**Contribution:** 2
**Rating:** 6
**Confidence:** 4

**Summary:**

The authors propose a new attribution evaluation framework which attempts to solve existing issues in current metrics such as the OOD problem from perturbation metrics, and the information leakage of fine-tuning methods like ROAR. To do this, the authors propose limiting the size of the perturbations made to each image and performing fine-tuning on images randomly masked under this perturbation size limitation.

**Strengths:**

From their experiments which cover a broad range of applications and datasets, it seems this new method does improve over existing methods in rank correlation tests, indicating its ability to fairly and consistently sort attributions under evaluation.

The solutions presented are simple.

**Weaknesses:**

I do not think the intuition behind the solutions is entirely clear and I do not think the presentation of the solutions is clear.

The choices for multiple hypermeters are not indicated.

More attribution methods should have been employed for a better representation of the existing space.

**Questions:**

The ablation study provided regarding B selection suggests this value can approach 1 (i.e. this is the same as if the value was not implemented) and the correlation scores would be good. Is this true? If so, what use does B have? In addition, what value of B is used and how was it selected?

What is the intuition for fine tuning with the image perturbation limitation? I do not see why fine-tuning under image perturbed by random masks instead of perturbed explanations is better either. It is not obvious that this skirts the OOD problem.

What is the value of being able to extract the explanation size? The motivation for this is lacking and it is not clear why it is important or interesting.

**Details Of Ethics Concerns:**

I realized that I accidentally pasted my question in this section. There are no ethics concerns.

---

> ### Author Response · Authors · 2024-11-22
> **Response to Reviewer S4br (Part 1/n)**
>
> Dear Reviewer S4br,
>
> Thank you very much for your time and valuable feedback on our manuscript. We sincerely appreciate your thoughtful comments and suggestions, which have greatly contributed to improving the quality of our paper. Here is our reply.
>
> >W1. I do not think the intuition behind the solutions is entirely clear and I do not think the presentation of the solutions is clear.
>
> **Problem Background:** In explainable AI, we evaluate explanations by removing the "important" parts of an input and measuring how this affects the model's prediction. However, when we remove these parts, the modified input may become "out-of-distribution" (OOD), making the model's predictions unreliable.
> **Existing Solutions and Their Limitations:** ROAR tries to solve this by retraining the model on inputs with explanations removed, but this leads to information leakage. R-Fidelity removes only portions of the explanation and averages over multiple runs, but this approach fails when the model is sensitive to small input changes or when the explanation covers a large part of the input.
> **Our Solution (F-Fidelity):** Our solution addresses these challenges through two key components. First, we fine-tune the model using random masking to make it robust to missing information. Second, during evaluation, we set an upper limit on how much of the input can be removed, preventing extreme domain shifts.
>
> To address the reviewer's concern, we have revised Section 3 to present these ideas more clearly, ensuring the strategies are presented in the same order as introduced. We have also added more detailed explanations of each component to improve clarity.
>
> > W2. The choices for multiple hypermeters are not indicated.
>
> Thank you for raising this question about hyperparameter choices. We have expanded our discussion of hyperparameters in several sections of the paper:
>
> In **Appendix C**, we now provide detailed specifications of our hyperparameters:
> - The fine-tuning and evaluation parameter `β` is set to 0.1 by default
> - For the number of sampling iterations `N`, we chose 50 as a balance between reducing variance and maintaining reasonable computation time
> - The OOD-mitigation parameters `α+` and `α-` are both set to 0.5
>
> To validate these choices, we conducted extensive sensitivity analyses, now available in **Appendix D.9**. These studies demonstrate that F-Fidelity maintains robust performance across a range of hyperparameter values. We provide comprehensive results of these analyses in **Appendix E.4**.
>
> Additionally, we have added a new section (**Appendix D.3**) specifically investigating the effects of `β`, since this parameter is crucial for both fine-tuning and evaluation stages of our method.
>
> > W3. More attribution methods should have been employed for a better representation of the existing space.
>
> Thank you for the insightful suggestion. We have expanded our evaluation to include more attribution methods:
>
> In addition to our original experiments with Integrated Gradients, SmoothGrad Square, and GradCAM, we have now conducted new experiments using LayerCAM. These additional experiments follow the same setup as Table 1, using a ResNet architecture on the CIFAR-100 dataset. The complete results are available in **Appendix D.5** and **Table 10**.
>
> The new results with LayerCAM show consistent patterns with our previous findings. F-Fidelity continues to demonstrate superior performance across multiple evaluation metrics. These additional experiments further validate the robustness of our evaluation framework across different explanation methods. We would be happy to include more attribution methods if specific ones are of particular interest.
>
> > Q1. The ablation study provided regarding B selection suggests this value can approach 1 (i.e. this is the same as if the value was not implemented) and the correlation scores would be good. Is this true? If so, what use does B have? In addition, what value of B is used and how was it selected?
>
>
> Thank you for these insightful questions about the role and selection of `β`. There appears to be a misunderstanding we'd like to clarify:
>
> Our experiments tested β values in the range [0.05, 0.25], not approaching 1.0. We have now documented in **Appendix C** that we use β = 0.1 by default. This choice reflects a careful balance of two factors:
>
> 1. If β is too small, the model won't learn sufficient robustness to missing information
> 2. If β is too large, the fine-tuned model may deviate significantly from the original model's decision boundaries
>
> To provide a complete understanding of β's impact, we have added a comprehensive analysis in **Appendix D.3**. This study shows that while our method is relatively robust to β selection within a reasonable range.

---

> ### Author Response · Authors · 2024-11-22
> **Response to Reviewer S4br (Part 2/n)**
>
> > Q2. What is the intuition for fine tuning with the image perturbation limitation? I do not see why fine-tuning under image perturbed by random masks instead of perturbed explanations is better either. It is not obvious that this skirts the OOD problem.
>
>
> Thank you for this insightful question about the foundations of our fine-tuning approach. The intuition behind our method can be explained in two parts:
>
> **Why limit perturbations during fine-tuning?**
> The goal is to make the model robust to missing information while preserving its original decision-making behavior. Large perturbations could fundamentally alter the model's learned patterns, while controlled perturbations help the model learn to handle missing information without drastically changing its decision boundaries.
>
> **Why use random masks instead of explanation-based perturbations?**
> Random masking provides several advantages:
> - Avoids information leakage that occurs when using explanation-guided perturbations (as in ROAR)
> - Creates a uniform distribution of modified inputs, helping the model learn general robustness to missing information
> - Ensures the fine-tuning process is explanation-agnostic, allowing fair comparison between different explanation methods
>
> By limiting perturbation size and using random masks, we create a controlled environment where the model becomes robust to missing information while staying close to its original distribution. This allows us to more accurately assess whether removed features truly contributed to model predictions, rather than measuring the model's sensitivity to distribution shifts.
>
>
> > Q3. What is the value of being able to extract the explanation size? The motivation for this is lacking and it is not clear why it is important or interesting.
>
> Thank you for asking about the significance of extracting explanation size. This capability offers several important practical benefits. First, traditional methods like Fidelity and ROAR require evaluating explanations across many sparsity levels, which is computationally expensive. Identifying the optimal sparsity point (explanation size) can lead to more efficient evaluation.
> Knowing the true explanation size helps in conducting fair comparisons between different explanation methods at the most relevant sparsity level. It enables validation of explanation methods on datasets where ground truth explanation sizes are known. Additionally, it provides valuable insights into model behavior and decision-making processes.
>
> ___
> We really appreciate the reviewer's time and efforts in reviewing the paper and providing valuable comments.  We have made substantial revisions to address each concern. We believe these changes have significantly strengthened the paper and improved its clarity. We are grateful for this opportunity to enhance our work and would be happy to address any additional questions or concerns you may have.

---

### Official Review · Reviewer_NNY1 · 2024-11-03

**Soundness:** 2
**Presentation:** 1
**Contribution:** 2
**Rating:** 8
**Confidence:** 3

**Summary:**

The authors propose a fidelity evaluation framework which utilizes an explanation agnostic fine tuning and a random mask generator which ensures that the generates input is not OOD. The efficacy of the proposed method if shown in various data types like images, time series etc.

**Strengths:**

1) The paper deals with important aspect of fidelity evaluation
2) The authors discuss about OOD issue during fidelity evaluation
3) The authors propose an approach to handle the issues and show the performance in different data types like images, time series and NLP.

**Weaknesses:**

W1) In section 3, it is mentioned in lines [215 to 222] that the size of the removed part is upper bounded by a fraction of the input and not a fraction of the explanation. It would be helpful to the readers if the authors could explain what they mean by "removed," as it might mean different in tabular data and images. Further, the authors should also explain how their strategy of selecting a fraction of input and not explanation addresses the OOD issue.

W2) In lines[224:235], the authors use fine tuning to address the problem of robustness of the classifier for small perturbations(mentioned in lines [213:214]). However, at least from the perspective of images, it is unclear how applying a mask of {0,1} on random patches of an image can be considered as a perturbation.

W3) Further, the fine-tuning process mentioned in lines[224:235] seems like retraining the classifier with occlusion (for image classifiers). The authors should explain how their method won't change the decision boundary drastically so that their approach can be called fine-tuning and not training a different classifier altogether.

W4) The authors should explain why explaining the fine-tuned classifier is equivalent to explaining the original classifier (as mentioned in line[224])

W5) For the ease of readers, the authors should explain more about GT and how they extract it(in Section 4)

W6) It needs to be clarified to see why F-Fidelity performs worse than vanilla Fidelity and R-Fidelity for SST2 dataset with LSTM (Table 8). The authors are requested to explain this phenomenon in detail for the ease of readers.

**Questions:**

Questions are as below

#### W1:
- a) Provide specific examples of how "removal" is implemented for different data types like images, tabular data, and text.
- b) Give a more detailed explanation of how bounding the removed portion to a fraction of the input, rather than the explanation, helps mitigate OOD issues.

#### W2:
- a) Elaborate on how the random masking process relates to or simulates small perturbations, particularly in the context of image data.
- b) Provide examples or visualizations that demonstrate how this masking approach compares to traditional perturbation methods.
- c) Present empirical evidence or theoretical justification for why their masking approach is an effective way to improve classifier robustness.

#### W3:
- a) Provide empirical evidence or theoretical justification for why their fine-tuning process preserves the original classifier's decision boundaries.
- b) Compare the original and fine-tuned classifiers' outputs on a test set to demonstrate the degree of change.

#### W4:
- a) Provide theoretical or empirical evidence demonstrating the equivalence between explanations of the original and fine-tuned classifiers.
- b) Include a comparison of explanations generated for both classifiers on a set of test examples.

#### W5:
- a) Provide a step-by-step description of how they generate or obtain the ground truth (GT) explanations, including any assumptions made in this process.
- b) Include examples of GT explanations for different data types.

#### W6:
- a) Provide a detailed analysis of why F-Fidelity underperforms on this specific dataset and model combination.
- b) Investigate potential reasons, such as dataset characteristics, model architecture specifics, or hyperparameter settings that might contribute to this result.
- c) Discuss the implications of this finding for the broader applicability of their method.
- d) Consider conducting an ablation study to isolate the factors contributing to this performance difference.

The paper addresses an important aspect of fidelity evaluation and the issue of OOD while doing it. Overall, I like the central theme of the paper and that’s why I am leaning towards mild acceptance but I would like the authors to address the points in the weakness section and the questions in this section.

---

> ### Author Response · Authors · 2024-11-22
> **Response to Reviewer NNY1(Part 1/n)**
>
> Dear Reviewer NNY1:
>
> Thank you very much for taking the time to review our manuscript and for providing insightful and valuable comments. Your feedback has been incredibly helpful in improving the quality of this work. We provide the reply in the following.
>
> >W1. In section 3, it is mentioned in lines [215 to 222] that the size of the removed part is upper bounded by a fraction of the input and not a fraction of the explanation. It would be helpful to the readers if the authors could explain what they mean by "removed," as it might mean different in tabular data and images. Further, the authors should also explain how their strategy of selecting a fraction of input and not explanation addresses the OOD issue..
> >W1 \& Q1.a. The mean and Specific examples of "removal" and how it is implemented for different data types like images, tabular data, and text.
>
> Thanks for providing this question. The "removal" means replacing the attribution with a constant value, which is a common operation in most explainable AI papers, including [1] and [2]. We have added concrete examples of "removal" operations in the introduction: for images, removal means setting pixels to zero/black; for time series, it means masking feature values with zeros; for text, it means replacing tokens with [MASK] token or zero embeddings.
>
>     - [1].Hooker, Sara, et al. A benchmark for interpretability methods in deep neural networks. NeurIPS, 2019.
>     - [2].Rong, Yao, et al. A consistent and efficient evaluation strategy for attribution methods. ICML, 2022.
>
> > b) Give a more detailed explanation of how bounding the removed portion to a fraction of the input, rather than the explanation, helps mitigate OOD issues.
>
> We thank the reviewer for this important question. Let us explain how our input-fraction-based removal strategy addresses OOD issues:
>
> First, during fine-tuning, our random masking approach helps the model learn to handle missing information consistently. By limiting removals to a fixed fraction `β` of the input size, rather than a fraction of the explanation size, we ensure the model specifically adapts to a controlled level of perturbation. This is crucial because explanation sizes can vary widely - for example, an explanation might cover 80% of an input, but removing that much information would likely create severe OOD issues.
>
> We have added empirical validation in **Appendix D.2** and **D.3**. The newly added **Table 8** shows that with ResNet on CIFAR-100, while the original model's accuracy drops from 73.23% to 41.81% with 10% masking, our fine-tuned model maintains consistent performance (73.10% to 72.97%).
>
> Second, explanation-based retraining approaches like ROAR face two key limitations: information leakage when retraining with explanation outputs, and computational inefficiency from requiring retraining for each explainer. Our explanation-agnostic fine-tuning addresses both issues while maintaining robust performance under perturbations.
>
>
> > W2) In lines[224:235], the authors use fine tuning to address the problem of robustness of the classifier for small perturbations(mentioned in lines [213:214]). However, at least from the perspective of images, it is unclear how applying a mask of {0,1} on random patches of an image can be considered as a perturbation.
> > a) Elaborate on how the random masking process relates to or simulates small perturbations, particularly in the context of image data.
>
> Thanks for this valuable question. Binary masking is widely used in influential perturbation-based XAI works. For example, LIME segments images into superpixels and masks them to understand model behavior[3]. Our approach builds upon this established methodology. Intuitively, binary masking effectively sets pixel values to zero, which represents the removal of information from those regions - this directly measures how the removal of certain input elements affects model predictions.
>
> [3] Ribeiro, Marco Tulio, Sameer Singh, and Carlos Guestrin. "" Why should I trust you?" Explaining the predictions of any classifier." Proceedings of the 22nd ACM SIGKDD international conference on knowledge discovery and data mining. 2016.

---

> > ### Comment · Reviewer_NNY1 · 2024-11-25
> > **Comment to Authors**
> >
> > I thank the authors for their explanations of my questions. I am satisfied with it and the experiments. I have a suggestion for future works regarding perturbation for images. Although LIME sets all pixels of segments to 0/1 to simulate removal, it might lead to OOD issues. This was noted by a recent work that saw higher fluctuation in the output probability using LIME's perturbation strategy as compared to using Gaussian Blur with different values [Bora et al., CVPR 2024]. It would be interesting to see how the proposed method works with Gaussian Blur rather than 0/1 for images.

---

> ### Author Response · Authors · 2024-11-22
> **Response to Reviewer NNY1(Part 2/n)**
>
> > W2 & Q2.b. Provide examples or visualizations that demonstrate how this masking approach compares to traditional perturbation methods.
>
> Thanks for the suggestions. We appreciate this comment but want to clarify that binary masking is indeed the traditional and widely used perturbation method in XAI literature. For example, LIME perturbs an image by masking superpixels to analyze the model's behavior[3], and SHAP uses feature perturbation (commonly binary feature masking) in combination with a game-theoretic approach to estimate the contribution of individual features to a model's prediction[4]. Binary masking is preferred in XAI because it clearly demonstrates feature contribution (present vs. absent) and directly tests feature importance through removal, making the results more interpretable compared to continuous perturbations like adding noise or blurring. Our approach follows this established practice.
>
> To further address the reviewer's concerns, we have added visualization examples in **Appendix D.4**. These visualizations show attribution maps generated by applying  GradCAM in both the original and fine-tuned models (with random mask), demonstrating that while our fine-tuning process improves model robustness to perturbations, it preserves the model's attention to meaningful features.
>
> [3] Ribeiro, Marco Tulio, Sameer Singh, and Carlos Guestrin. "" Why should I trust you?" Explaining the predictions of any classifier." Proceedings of the 22nd ACM SIGKDD international conference on knowledge discovery and data mining. 2016.
> [4] Lundberg, Scott. "A unified approach to interpreting model predictions." NeurIPS (2017).
>
>
> > W2.& Q2.c) Present empirical evidence or theoretical justification for why their masking approach is an effective way to improve classifier robustness.
>
> We sincerely thank the reviewer for this insightful suggestion. Following the reviewer's guidance, we have added a comprehensive analysis of our masking approach's effectiveness in **Appendix D.2**. The analysis compares classification accuracy between original and fine-tuned models under different masking ratios (0%, 5%, 10%) across three domains:
>
> In **computer vision** (Cifar-100, ResNet), our fine-tuned model maintains consistent accuracy (~73%) even with 10% masking, while the original model's accuracy drops significantly from 73.23% to 41.81%.
>
> In **time series** (Boiler dataset, LSTM), the fine-tuned model shows improved robustness, maintaining 73.19% accuracy under 10% masking compared to the original model's drop to 61.53%.
>
> In **NLP** (SST2, LSTM), both original and fine-tuned models maintain similar high accuracy (~79%) even with 10% masking, revealing that NLP models are inherently robust to masking perturbations. This explains why traditional fidelity metrics perform similarly to our method on NLP tasks.
>
> These patterns demonstrate that our fine-tuning strategy effectively improves model robustness where needed, particularly in computer vision and time series domains where models are naturally more sensitive to input perturbations. We have included detailed results for additional architectures (Transformer, CNN) in **Table 8** of the appendix.
>
> We appreciate the reviewer's suggestion as it has helped us better validate and present the effectiveness of our approach.
>
> | **Table 8**  | Perturbation Ratio | 0% | 5% | 10% |
> |---|-------------------|-----|-----|-----|
> | Cifar-100|  | | | |
> |  Resnet  | Original Model    | 73.23% | 56.70% | 41.81% |
> |          | Fine-tuned Model  | 73.10% | 73.12% | 72.97% |
> |  Transformer  | Original Model| 50.25% | 46.19% | 39.87% |
> |          | Fine-tuned Model  | 46.91% | 47.05% | 47.30% |
> | Boiler   |  | | | |
> | LSTM     | Original Model    | 80.14% | 67.78% | 61.53% |
> |          | Fine-tuned Model  | 77.22% | 72.64% | 73.19% |
> | CNN     | Original Model     | 82.22% | 68.89% | 57.92% |
> |          | Fine-tuned Model  | 86.39% | 74.03% | 63.89% |
> | SST-2    |  | | | |
> | LSTM     | Original Model    | 82.45% | 80.28% | 79.82% |
> |          | Fine-tuned Model  | 82.68% | 80.16% | 79.24% |
> |Transformer| Original Model   | 80.85% | 77.29% | 73.28% |
> |          | Fine-tuned Model  | 80.73% | 81.08% | 78.90% |

---

> ### Author Response · Authors · 2024-11-22
> **Response to Reviewer NNY1(Part 3/n)**
>
> >W3) Further, the fine-tuning process mentioned in lines[224:235] seems like retraining the classifier with occlusion (for image classifiers). The authors should explain how their method won't change the decision boundary drastically so that their approach can be called fine-tuning and not training a different classifier altogether.
> W3 & Q3.a. The authors should explain how their method won't change the decision boundary drastically so that their approach can be called fine-tuning and not training a different classifier altogether. Provide empirical evidence or theoretical justification for why their fine-tuning process preserves the original classifier's decision boundaries
>
> Thank you very much for these insightful comments. Let us first interpret the reviewer's concern and share our thoughts.
>
> We use the term "fine-tuning" because we continue training from a pre-trained model, rather than training from random initialization. The reviewer raises an important concern: if the original model and fine-tuned model have different decision boundaries, then explaining the fine-tuned model may not accurately reflect the behavior of the original one.
>
> We have added a new section in **Appendix D.3** to comprehensively investigate this problem. Specifically, we introduce two intuitive concepts: Global Decision Boundary and Local Decision Boundary. The global one refers to the overall separation between different classes in the input space. It captures the main structure of the data distribution and represents the classifier's ability to distinguish between classes on a broad scale. The local one characterizes how the model responds to small perturbations around individual data points.
>
> Our random masking fine-tuning aims to maintain the model's global classification behavior while improving its stability to small input changes. The random masking process applies uniform noise across all inputs, regardless of their class. This helps the model learn to handle missing information while keeping its main classification patterns intact.
>
> To empirically evaluate our fine-tuning approach, we conduct comprehensive experiments with ResNet on CIFAR-100. As shown in **Figure 1**, our analysis examines both global decision boundary preservation and local boundary characteristics through systematic perturbation analysis. For global boundary preservation, we compare classification accuracy between the original and fine-tuned models on clean test data, using 0% masking as the baseline reference point. We also measure prediction agreement between these models to assess consistency in their decision-making.
>
> Our results in **Figure 1** demonstrate that random masking fine-tuning effectively balances global decision boundary preservation with local boundary smoothing. The fine-tuned models maintain comparable baseline accuracy to the original model while showing significantly improved robustness to perturbations. Figure 1 visualizes this trade-off: models fine-tuned with larger β values demonstrate remarkable resilience, maintaining accuracy above 60% even under 50% input perturbation, while the original model drops below 40% accuracy with just 10% perturbation. However, higher β values also lead to lower agreement with the original model's predictions, suggesting that stronger fine-tuning causes the model to learn more robust but slightly different decision boundaries. Models with moderate β values (like 0.10) maintain a better balance between perturbation resistance and consistency with the original model's behavior.
>
> We further validate this through our comprehensive experimental design. Our evaluation methodology uses controlled degradation of explanations: we start with a well-established explainer to generate good explanations of the original model (which can also be visually verified), then systematically create explanations of known quality by mixing the original explanation with different levels of random noise. Then, we compare evaluation metrics with the ground truth ranking. Note that this experiment setting is not affected by the mentioned "inconsistency behavior problem" because our ground truth ranking is based on the noise level in explanations of the original model. Our results show that the fine-tuned model preserves meaningful explanation evaluation by consistently producing rankings that match these known ground truth quality levels.

---

> ### Author Response · Authors · 2024-11-22
> **Response to Reviewer NNY1(Part 4/n)**
>
> > W3 & Q3.b. Compare the original and fine-tuned classifiers' outputs on a test set to demonstrate the degree of change
>
> Thank you for the valuable suggestion. We have conducted comprehensive experiments comparing the original and fine-tuned classifiers' performance on test sets across multiple datasets and architectures. The detailed results have been added to **Appendix D.3**.
>
> For CIFAR-100 with ResNet backbone, we measure both classification accuracy and prediction agreement under various masking ratios from 0% to 50%. With our recommended fine-tuning parameter β=0.1, the results show:
>
> On clean data (0% masking), the fine-tuned model achieves 73.1% accuracy compared to 73.2% for the original model, demonstrating strong preservation of global decision boundaries. The high prediction agreement between models on clean data further confirms this preservation. Under increasing masking ratios, the original model's accuracy drops sharply to 41.8% at 10% masking, while the fine-tuned model maintains 73.0% accuracy, showing improved robustness while preserving the original model's clean-data behavior.
>
>
> > W4 & Q4.a The authors should explain why explaining the fine-tuned classifier is equivalent to explaining the original classifier (as mentioned in line[224]). Provide theoretical or empirical evidence demonstrating the equivalence between explanations of the original and fine-tuned classifiers
>
> Thank you for raising this important point about the relationship between explanations of the original and fine-tuned classifiers. We appreciate the opportunity to clarify this aspect of our work.
> From a theoretical perspective, our random masking fine-tuning is designed to preserve the model's core decision-making patterns while only improving its resilience to missing information. The random nature of the masking process ensures that the model learns to handle perturbations without biasing toward specific features or classes.
>
> Empirically, we have added two sets of experiments to validate this consistency. In **Appendix D.3**, we demonstrate that the fine-tuned model maintains similar predictive behavior to the original model on clean inputs, achieving comparable accuracy (73.1% vs 73.2% on CIFAR-100) with reasonable prediction agreement on the test set.
>
> Furthermore, we added qualitative evidence in **Appendix D.4**, where we visualize and compare attribution maps generated by GradCAM for both models on Tiny-Imagenet. As shown in **Table 9**, both models highlight similar regions of importance, with only minor variations in highlight intensity. This visual consistency demonstrates that our fine-tuning process improves robustness while preserving the model's attention to meaningful features.
>
> - Q4.b Include a comparison of explanations generated for both classifiers on a set of test examples
>
> Thank you for this helpful suggestion. We have added a new section **Appendix D.4** to provide a detailed qualitative comparison between explanations generated for both classifiers.
>
>
> > W5 & Q5.a For the ease of readers, the authors should explain more about GT and how they extract it(in Section 4). Provide a step-by-step description of how they generate or obtain the ground truth (GT) explanations, including any assumptions made in this process.
>
> Thank you for pointing out this need for clarification. We respectfully believe there may be a misinterpretation of our experimental setting. In Section 4, rather than generating ground truth (GT) explanations, we create ground truth rankings through a controlled experimental setup.
>
> Our approach begins with explanations from well-established explainers (e.g., Integrated Gradient, GradCAM) that have been extensively validated in practice. We have also manually verified their high explanation quality and provide some examples in **Table 9**. So we assume that they are good explanations.
>
> We then systematically degrade these explanations by adding different ratios of random noise [0.0, 0.2, 0.4, 0.6, 0.8, 1.0]. This creates a sequence of explanations with known relative quality - explanations with less noise should naturally rank higher than those with more noise. This controlled setup provides a ground truth ranking that allows us to rigorously verify whether evaluation metrics can correctly recover the known ordering of explanation quality.
>
> Based on your insightful suggestion, we have revised **Section 4** to better explain this controlled experimental design and its rationale. We appreciate your feedback as it helps us improve the clarity of our presentation.
>
> > Q5.b Include examples of GT explanations for different data types.
>
> Thanks for this question.  in Section 4, we introduce ground truth ranking instead of GT explanations. We have revised this part to make it clearer.

---

> ### Author Response · Authors · 2024-11-22
> **Response to Reviewer NNY1(Part 5/n)**
>
> > W6  It needs to be clarified to see why F-Fidelity performs worse than vanilla Fidelity and R-Fidelity for SST2 dataset with LSTM (Table 8). The authors are requested to explain this phenomenon in detail for the ease of readers.
> >> Q6.a.Provide a detailed analysis of why F-Fidelity underperforms on this specific dataset and model combination.
> >> Q6.b. Investigate potential reasons, such as dataset characteristics, model architecture specifics, or hyperparameter settings that might contribute to this result.
> >> Q6.c. Discuss the implications of this finding for the broader applicability of their method
> >> Q6.d. Consider conducting an ablation study to isolate the factors contributing to this performance difference
>
> We sincerely appreciate these insightful questions that help identify an important aspect of our work. Our investigation reveals an interesting relationship between model robustness and the effectiveness of different evaluation metrics.
> In our new analysis (**Appendix D.2**), we examine model behavior across computer vision (Cifar-100), time series (Boiler), and natural language processing (SST2) domains by measuring classification accuracy under different perturbation levels (0%, 5%, and 10%). For each, we consider both representative models. The results reveal distinct patterns across domains.
>
> Computer vision and time series models show high sensitivity to perturbations, with significant accuracy drops when perturbed. In these cases, F-Fidelity significantly outperforms traditional metrics by addressing the OOD problem through fine-tuning.
> However, NLP models, particularly on SST2 with LSTM, demonstrate remarkable inherent robustness - maintaining around 80% accuracy even with 10% perturbation (**Table 8**). In such naturally robust scenarios, the OOD problem that F-Fidelity addresses is less severe, explaining why simpler metrics perform adequately.
>
> These findings have important practical implications: F-Fidelity should be preferred for domains where models show high sensitivity to perturbations, while simpler fidelity metrics may suffice for naturally robust domains. Based on these insights, we have added practical guidelines for metric selection in **Section 4.4** of the main paper.
>
> We are grateful for the reviewer's questions which led us to discover these valuable insights about the relationship between model robustness and evaluation metric selection.
>
> ____
> We sincerely thank the reviewer for your time and thoughtful feedback. Your comments have helped us identify important areas for clarification and improvement in our paper. We have made substantial revisions to address each concern, including adding detailed analyses, clarifying our methodology, and providing practical guidelines for using our method.
>
> We believe these changes have significantly strengthened the paper and improved its clarity. We are grateful for this opportunity to enhance our work and would be happy to address any additional questions or concerns you may have.

---

> ### Author Response · Authors · 2024-11-26
> **Thank You to Reviewer NNY1**
>
> We sincerely thank the reviewer for the thoughtful questions and detailed suggestions, which have greatly contributed to enhancing the clarity and rigor of our work. We are pleased that our explanations and experiments have addressed your concerns. We greatly appreciate your insightful suggestion regarding the use of Gaussian Blur as an alternative perturbation strategy. We have updated the conclusion to include this as a direction for future work. Thank you again for your valuable feedback and constructive suggestions, which have significantly enriched our work.

---

### Author Response · Authors · 2024-11-22
**Global Response to Reviewers**

Dear Reviewers,

We sincerely appreciate your valuable feedback on our manuscript. In response to these comments, we have made substantial revisions to enhance the paper's clarity and empirical validation. Below is a detailed summary of our revisions.

- We have reorganized the paper to improve its flow and readability. Our primary contribution - a novel evaluation framework for measuring explanation faithfulness that is theoretically grounded and robust to distribution shifts - is now more clearly presented.

- Regarding methodology, we have revised the description to clearly present our method. We summarize our strategy as **fine-tuning and bounded stochastic removal**. That is we propose an explanation-agnostic fine-tuning strategy to enhance model robustness and introduce a removal bound to ensure in-distribution evaluation inputs.

- Our experimental validation has been considerably strengthened through testing across multiple data modalities, including images, time series, and natural language. We have included additional case studies demonstrating F-Fidelity's superior performance in Appendices D.6 and D.10, supported by extensive analysis of model robustness to perturbations in Appendix D.2. Based on our empirical findings, we have provided implementation guidelines in Section 4.4. A detailed analysis of changes in the model's decision boundary post-fine-tuning is provided in Appendix D.3, along with comparative case studies visualizing explanations from both original and fine-tuned models.

- To address concerns about parameter selection, we have expanded our discussion in Appendix E.4 and added new analysis in Appendix D.3 explaining parameter choices. We demonstrate our method's robustness to hyperparameter variations and provide detailed justification for the selection of β, which balances robustness and decision boundary stability.

- Regarding computational efficiency, we have included a comprehensive comparison of computational costs between ROAR and F-Fidelity in Appendix D.11. This analysis highlights the significant speed advantages offered by our explanation-agnostic fine-tuning strategy.

These revisions collectively strengthen the practical utility and empirical validation of our F-Fidelity framework. We believe the manuscript now presents a more robust and clearer contribution to the field.

We are grateful for your valuable feedback, which has helped substantially improve our work. Thank you for your time and consideration.

---

> ### Comment · Reviewer_S4br · 2024-11-24
>
> I would like to thank the authors for their diligence in responding to both mine and the other reviewer's concerns. I believe many of these additional experiments have greatly improved the persuasiveness of this work's effectiveness. These results have led me to changing my score to a marginal accept. I think this is a worthy competitor to the ROAR method not only in performance, but also the reduced computational requirements.

---

> ### Author Response · Authors · 2024-11-26
>
> We sincerely thank the reviewers for their insightful feedback and acknowledgment of our efforts to address the concerns raised.  We appreciate your recognition of the method's advantages compared to ROAR, including its effectiveness and reduced computational requirements. Thank you again for your constructive comments, which have contributed to significantly enhancing the clarity of this work.

---

### Meta-Review · Area_Chair_NrRX · 2024-12-21

**Metareview:**

Strengths :
The paper addresses the important issue of out-of-distribution (OOD) inputs in fidelity evaluation and proposes a simple yet effective method to handle this challenge. Through extensive experiments across diverse data types, including images, time series, and NLP, the authors demonstrate that their approach improves rank correlation in attribution evaluations, ensuring more reliable and consistent assessments. The method effectively mitigates limitations of prior techniques by addressing both information leakage and OOD problems.

Weaknesses:
It lacks visualizations, such as attribution maps, to demonstrate the effectiveness of F-Fidelity and highlight its advantages over other evaluation metrics. The novelty of F-Fidelity is questioned, as its primary distinction from R-Fidelity seems incremental. The use of Integrated Gradients for establishing ground truth raises concerns, as it may not be the most faithful method compared to others like LRP or LayerCAM. The importance of macro correlation is not sufficiently explained, and the paper evaluates a limited set of explanation methods, missing comparisons with more established techniques. Additionally, practical challenges in determining the size of the most influential tier (c₁) and clarifications on certain aspects of the methodology (e.g., perturbation masking and fine-tuning) are not fully addressed. There are also issues with the clarity of the intuition behind the solutions, hyperparameter choices, and the paper's presentation of the results.

**Additional Comments On Reviewer Discussion:**

There was a long discussion among the authors and reviewers and most reviewers had most of their concerns addressed and some raised their rating. In the end, all ratings are positive so will be accepted.

---

### Decision · Program_Chairs · 2025-01-22

Accept (Poster)